

# One year monitoring of volatile organic compounds (VOCs) from an oil-gas station in northwest China

Huang Zheng [1,2], Shaofei Kong [2], Xinli Xing [1,3], Yao Mao [3], Tianpeng Hu [1], Yang Ding [1], Gang Li [4], Dantong Liu [5], Shuanglin Li [2], and Shihua Qi [1,3]

[1] Department of Environmental Science and Technology, School of Environmental Studies, China University of Geosciences, Wuhan, 430074, China
[2] Department of Atmospheric Sciences, School of Environmental Studies, China University of Geosciences, Wuhan, 430074, China
[3] State Key Laboratory of Biogeology and Environmental Geology, China University of Geosciences, Wuhan, 430074, China
[4] Karamay Environmental Monitoring Center Station, Karamay, 834000, China
[5] School of Earth and Environmental Sciences, the University of Manchester, M13 9PL, UK

*Correspondence to*: Shaofei Kong (kongshaofei@cug.edu.cn) or Xinli Xing (xingxinli5300225@163.com)

**Abstract.** Oil and natural gas are important energy supply around the world. The exploring, drilling, transportation, and processing in oil-gas regions can release abundant volatile organic compounds (VOCs). To understand the atmospheric behaviors of VOCs in such region, the fifty-six VOCs designed as the photochemical precursors by the United State Environmental Protection Agency were continuously measured for an entire year (September 2014-August 2015) by a set of on-line monitor system at an oil-gas station in northwest China. The VOC concentrations in this study were 1-50 times higher than those measured in many other urban and industrial regions. The VOC compositions were also different from other studies with alkanes contributing up to 87.5% of the total VOCs in this study. According to the propylene-equivalent concentration and maximum incremental reactivity method, alkanes were identified as the most important VOC groups to the ozone formation potential. The photochemical reaction, meteorological parameters (temperature, relative humidity, pressure, and wind speed) and boundary layer height were found to influence the temporal variations of VOCs at different time scales. The positive matrix factorization analysis showed that the natural gas, fuel evaporation, combustion sources, oil refining process, and asphalt contributed 62.6%, 21.5%, 10.9%, 3.8%, and 1.3%, respectively to the total VOCs on the annual average. Clear temporal variations differed from one source to another was observed, due to their differences in source emission strength and the influence of meteorological parameters. Potential source contribution function and contribution weighted trajectory models based on backward trajectories indicated that five identified sources had similar geographic origins. Raster analysis based on CWT analysis indicated that the local emissions contributed 48.4%-74.6% to the VOCs. This research filled the gaps in understanding the VOCs in the oil-gas field region, where exhibited different source emission behaviors compared with the urban/industrial regions.

***Keywords***: Volatile organic compounds; oil-gas field; spatial variation; photochemical behavior; source apportionment; source region and local-regional contribution





## 1 Introduction

Volatile organic compounds (VOCs) are ubiquitous in ambient air and originate from both natural processes (i.e., vegetation emission, volcanos eruption and forest fire) and anthropogenic activities, including fossil fuel combustion, industrial processes and solvent usage (Cai et al., 2010; Leuchner and Rappenglück, 2010; Baudic et al., 2016). As key precursors of the $O_3$ formation (Fujita, 2001; Geng et al., 2008; Ran et al., 2009; Lyu et al., 2016), VOCs also exhibit detrimental effects on human health (Colman Lerner et al., 2012; He et al., 2015) and air quality.

Previous studies mainly focused on measurements in urban agglomerations, such as the Pearl River Delta (PRD) region (Tang et al., 2007; Liu et al., 2008b; Cheng et al., 2010; Ling et al., 2011), Yangtze River Delta (YRD) region (An et al., 2014; Li et al., 2016; Shao et al., 2016) and Beijing-Tianjin-Hebei (BTH) region (Li et al., 2015) and megacities including Beijing (Song et al., 2007; Wang et al., 2010; Yuan et al., 2010), Shanghai (Cai et al., 2010; Wang, 2014), Guangzhou (Zou et al., 2015) and Wuhan (Lyu et al., 2016) or industrial areas (An et al., 2014; Wei et al., 2015; Shao et al., 2016). These studies found that vehicle emission and solvent usage contributed most to the ambient VOCs. A few studies were conducted in petrochemical industrial regions with intensive VOC emissions (Lin et al., 2004; Liu et al., 2008a; Wei et al., 2015; Jia et al., 2016; Mo et al., 2017). Studies concerning VOCs at regions of oil-gas exploiting have been reported (Buzcu-Guven and Fraser, 2008; Simpson et al., 2010; Rutter et al., 2015; Bari et al., 2016). The contributors of VOCs at petrochemical regions are quite different from urban regions and are mainly depended on the processing stages (Warneke et al., 2014). Leuchner and Rappenglück. (2010) found that natural gas/crude oil sources contributed most to the VOC emissions in Houston. Similar results were reported in the Northeast Colorado due to the oil and natural gas operations (Gilman et al., 2013). In previous studies, the ambient air was sampled for a few days (weeks) or at a certain season with low time-resolution. The diurnal, monthly, and seasonal variations were mostly overlooked, which prevented the understanding of the VOCs' temporal behaviors influenced by the real-time emissions, photochemical reactions, and meteorological parameters. Therefore, a long term monitoring with the high-time resolution of VOCs is desired (Baudic et al., 2016; Liu et al., 2016). It should be emphasized that in September of 2013, the VOCs control at petrochemical regions has been listed as one of the main objectives of the Action Plan of Atmospheric Pollution Control released by the central government of China, which proposed new requirements to conduct researches in this type of field.

To identify the sources of VOCs, receptor models including chemical mass balance (CMB), positive matrix factorization (PMF), and principal component analysis/absolute principal component scores (PCA/APCS) have been widely used (Guo et al., 2004; Rodolfo Sosa et al., 2009; An et al., 2014; Liu et al., 2016). Meanwhile, dispersion models including conditional probability function (CPF), backward trajectory, potential source contribution function (PSCF), and concentration weighted trajectory (CWT) are also employed to locate their potential source origins (Song et al., 2007; Chan et al., 2011; Liu et al., 2016). Receptor models can only give the source categories and contributions, while the dispersion model can explore the potential geographic source regions. Recently, the combination of these two types of models was developed to figure out the



origins of various air pollutant sources (Zhang et al., 2013a; Bressi et al., 2014; Chen et al., 2016). While these practices mainly focus on the atmospheric fine particles ($PM_{2.5}$), few have concerned the local and regional source contributions of VOCs.

In this study, an oil-gas field located in the northwest China was chosen as the study area to conduct a long time monitoring of VOCs with high-time resolution. The main objectives are to (1) compare VOC concentrations, compositions, and ozone

formation potential at this oil-gas station with other researches, (2) explore the relationships between VOC concentrations and metrological parameters in different time scale, (3) employ PMF model to identify the possible VOC sources, and (4) identify the local contributions and regional origins of VOCs based on the combination of PMF and dispersion models. This study is the first VOCs research with high-time resolution at the oil-gas field in China, which provides new information on the temporal distribution, photochemical properties, and local/regional contributions of VOCs and contributes to establish the control

measures of VOCs at this type of region.

## 2 Materials and methods

### 2.1 Site description

The study area (44.1-46.3°N and 84.7-86.0°E) is located in the northwest China and at the northwestern margin of Junggar Basin, which is an important oil and gas-bearing basin (Fig. 1a). The proven deposits of oil and natural gas are $2.41 \times 10^9$ t

and $1.97 \times 10^{11}$ m$^3$, respectively in this area. There are hundreds of oil and gas wells in this oil-gas field with an annual gas deliverability of $1.20 \times 10^{10}$ m$^3$. Additionally, 126 petrochemical plants spread across this area. It can be divided into two regions with oil gas operation/oil refinery in the north direction (Region 1) and petrochemical industry in the south direction (Region 2). These two regions are about 150 km in distance (Fig. 1b). Region 1 is abundant in oil gas resources and the main petrochemical factories are oil refineries and natural gas chemical plants. The main products include gasoline, diesel, asphalt,

and 1, 3-butadiene and the chemical process flow charts are shown in Fig. S1. Region 2 is a key petrochemical base, with the production capacity of oil and ethylene being $6 \times 10^6$ t/yr and $2.2 \times 10^5$ t/yr, respectively. The sampling site is located on the rooftop of a building (15m above the ground, 45.6°N, 85°E), about 11 km away from the southeast of the urban region. At the northeast of the sampling site, there are hundreds of oil and gas wells (Fig. 1c). The study area is located deeply secluded hinterland of Eurasia. The typical temperate continental arid desert climate results in high temperature in summer (27.9 °C)

and low temperature in winter (-15.4 °C). The sufficient solar radiation, little precipitation and the low humidity (43%-56%) result in high evaporation (>3000 mm) in this region.

### 2.2 Descriptions of instruments and QA/QC

From September 2014 to August 2015, continuous online sampling and measurement of fifty-six Photochemical Assessment Monitoring Stations (PAMS) VOCs were conducted at the sampling site. The ambient VOCs were sampled continuously and

analyzed automatically at 2 h interval with an online monitor system (TH-300B, Wuhan-Tianhong Instrument Co., Ltd, China).



The sampling and analysis procedures were described elsewhere (Lyu et al., 2016). Briefly, two-channels were initialed to sample and analyze VOCs separately. The water and carbon dioxide in the sampled air was removed at a cold trap maintaining at -80 ℃. $C_2$-$C_4$ VOCs were trapped with a PLOT column (diameter: 0.32 mm, thickness of membrane: 5 mm, and length: 60 m). $C_5$-$C_{12}$ were trapped with an empty capillary column at -150 ℃. After the pre-concentration, the VOCs were desorbed by

5     rapid heating to 120 ℃. $C_2$-$C_4$ VOCs were introduced into a gas chromatograph-flame ionization detector (GC-FID, Agilent 7820) and $C_5$-$C_{12}$ VOCs were introduced into a gas chromatograph-mass spectrometer detector (GC-MSD, Agilent 5975) for further analysis.

The target compounds involved 56 VOC species: alkanes (29), alkenes (10), alkynes (acetylene), and aromatics (16). The standard gases from PAMS were used for the equipment calibration and verification through the 5-point method every two

10    weeks (Lyu et al., 2016). The correlation coefficients of the calibration curves usually varied from 0.991 to 0.998. The detection limits were in the range of 0.04 to 0.12 ppbv (Table 1). The missing value was due to power failure or instrument maintenance and was not included in the data analysis.

## 2.3 PMF receptor model description

Positive matrix factorization (PMF) model has been widely employed for VOCs source apportionment (Buzcu-Guven and

15    Fraser, 2008; Leuchner and Rappenglück, 2010; Liu et al., 2016; Lyu et al., 2016). It decomposes a matrix of X (i ×j dimension) into factor contributions matrix G (i ×k dimensions) and factor profiles matrix F (k ×j dimensions) plus a residue matrix E (i ×j dimension):

$$x_{ij} = \sum_{k=1}^{p} g_{ik} f_{kj} + e_{ij} \tag{1}$$

where $x_{ij}$ is the concentration of $j$th VOC species measured in $i$th sample, $g_{ik}$ represents the contribution of the $k$th source to

20    the $i$th sample, $f_{kj}$ represents the mass fraction of the $j$th compound from the $k$th source, and $e_{ij}$ is the residual for each sample/species. The purpose of the PMF model is to find the minimum Q value with uncertainty ($u_{ij}$) introduced into model:

$$Q = \sum_{i=1}^{n} \sum_{j=1}^{m} \left[ \frac{x_{ij} - \sum_{k=1}^{p} g_{ik} f_{kj}}{u_{ij}} \right]^2 \tag{2}$$

EPA PMF 5.0 (US EPA, 2014) was employed for VOCs source apportionment and more detailed PMF operations were given in Appendix A.

## 25    2.4 Meteorological parameters and air pollutants

The 3-hours resolution meteorological parameters including atmospheric pressure (P), temperature (T), relative humidity (RH), wind speed (WS) and direction (WD) are shown in Fig. 2. The boundary layer height (BLH) was computed every 3 hours each day through the NOAA's READY Archived Meteorological website (http://www.ready.noaa.gov/READYamet.php). Significant differences ($p<0.01$) were found between the meteorological parameters in different seasons (autumn: Sep, Oct



and Nov; winter: Dec, Jan and Feb; spring: Mar, Apr and May; summer: Jun, Jul and Aug). During the sampling period, northwesterly and northeasterly winds prevailed (Fig.1c).

The hourly data of trace gases including CO, $NO_2$, and $O_3$ were synchronously measured with an enhanced trace level CO analyzer (Model 48$i$TL, Thermo Environmental Instruments (TEI) Inc.), a chemiluminescence trace level $NO_2$ analyzer (Model 42$i$TL, TEI), and an UV photometric $O_3$ analyzer (Model 49$i$, TEI), respectively (Lyu et al., 2016). These three trace gases and hourly dataset of ambient fine particles ($PM_{2.5}$), airborne particulate matter ($PM_{10}$) and $SO_2$ (Model 43$i$, TEI) were collected from the local environmental monitoring station. According to the ambient air quality standards-Ⅱ (GB/3095-2012), the main air pollutants were $PM_{10}$ and $PM_{2.5}$ in winter and $NO_2$ in autumn (Fig. S2).

### 2.5 Ozone formation potential

The VOC concentrations are not proportional to the ozone formation potential due to their wide ranges of photochemical reactivity with OH radicals. For example, the rate constant of ethane reacting with OH radicals is $0.27 \times 10^{12}$ cm$^3$ molecule$^{-1}$ s$^{-1}$ and the rate constant of propylene is $26.3 \times 10^{12}$ cm$^3$ molecule$^{-1}$ s$^{-1}$ at 298 K. Two methods including propylene-equivalent concentrations (Propy-Equiv) and the maximum incremental reactivity (MIR) were adopted to analyze the ozone formation potential of VOCs. Their calculation equations are expressed as follows (Atkinson and Arey, 2003; Zou et al., 2015):

$$C_{j,Propy-Equiv} = C_{j,C} \times \frac{k_{j,OH}}{k_{Propy,OH}} \qquad (3)$$

$$C_{j,MIR} = MIR_j \times C_{j,ppbv} \times \frac{m_j}{M} \qquad (4)$$

where $C_{j,C}$ and $C_{j,ppbv}$ represent the carbon atom concentration (ppbC) and actual concentration by volume (ppbv) for species $j$, respectively. $k_{j,OH}$, $k_{propy,OH}$ and $MIR_j$ denote the chemical reaction rate constant in the free radical reaction of species $j$, propylene with OH and maximum incremental reactivity for species $j$, respectively. $m_j$ and $M$ stand for the relative molecular mass of species $j$ and the molecular mass of ozone, respectively. The estimated MIR coefficients and the OH reaction rate constants from former studies are listed in Table 1.

### 2.6 Geographic origins of the VOCs
### 2.6.1 Conditional probability function (CPF)

The CPF is widely used to locate the direction of sources based on wind direction data (Song et al., 2007). In this study, the directions of various VOC sources were explored based on the $G$ matrix in PMF analysis and wind directions. The CPF is defined as Eq (5):

$$CPF = \frac{m_{\Delta\theta}}{n_{\Delta\theta}} \qquad (5)$$

where $m_{\Delta\theta}$ is the number of data from wind sector $\Delta\theta$ (each is 22.5 degree) that exceed the threshold value (75th percentile of each source contribution); $n_{\Delta\theta}$ is the total number of occurrence from the same wind direction. Calm conditions (wind speed< 1m/s) were excluded from the calculation for its difficulty in defining the wind direction.





### 2.6.2 Backward trajectory analysis

The 48 h backward trajectories at 2 h intervals (starting from 0:00 to 20:00 local time, LT) were run each day by the software-TrajStat (Wang et al., 2009; Squizzato and Masiol, 2015). The top of the model was set at 500 m above ground level (Zhao et al., 2015; Liu et al., 2016). The FNL global analysis data produced by the National Center for Environmental Prediction's

Global Data Assimilation System (GDAS) wind field re-analysis was introduced into the TrajStat model. A total of 2743 backward trajectories were calculated and then were grouped into four clusters according to their geographic sources and histories (Fig. S3). The trajectories were mainly originated from the northwest during the whole sampling period.

### 2.6.3 Local and regional transport contribution

The potential source contribution function ( PSCF ) and concentration weighted trajectory (CWT) model are previously used

to identify the possible source regions based on the backward trajectory analysis (Cheng et al., 2013; Bressi et al., 2014; Liu et al., 2016). The PSCF gives the proportion of air pollution trajectory in a given grid and the CWT reflects the concentration levels of trajectories. The geographic domain (31 °71 ′N, 36 °107 ′E) was found to be within the annual range of 48h backward trajectories. The total number of grids was 11360 with a resolution of 0.5 °×0.5 °. More information about the PSCF and CWT analysis can be found in Appendix B.

Local and regional source contributions of the observed VOCs were calculated by raster analysis. In previous studies, the domain was divided into 12 sectors (each was 30 ′) to study the regional contributions (Bari et al., 2003; Wang et al., 2015; Wang et al., 2016). However, in this study, the domain was briefly divided into two sections (local and regional), with the sampling site as the original point. The range of local sources was defined as a polar with a radius of 12 h backward trajectories and the range of regional sources was outside of the circle (detailed descriptions can be found in Appendix C). The

concentration of each grid was calculated by CWT analysis. By counting and averaging in each section, contributions of local emission and regional transportation were produced. To reduce the effects of background values, the lowest CWT values ($C_b$) in each section was deduced from the concentrations. The contribution (%) of local source and regional transportation was defined as follows:

$$\%C_i = \frac{(C_i - C_{bi}) \times N_i}{\sum_{i=1}^{2} (C_i - C_{bi}) \times N_i} \times 100\% \tag{6}$$

where $C_i$ is the mean CWT value in $i$th section (local or regional), $C_{bi}$ is the background value of the $i$th section, $N_i$ is the number of grid with non-zero CWT concentrations in $i$th section.

Several factors affect the calculated results of regional and local source contributions, including the radius of the circle and CWT value. In this study, the 12 h backward trajectories were chosen to differentiate the local area from regional area. However, the longer the backward trajectories were, the lower regional contributions produced. In addition, the PMF model

was employed to VOCs source apportionment and the contribution of each identified source was introduced into CWT calculation. However, the negative value of source contribution was inevitably generated despite the application of F-peak in PMF analysis. Therefore, the negative CWT value was excluded in raster analysis and this would affect the results of regional



and local source contributions. Overall, although flaws existed in this new method, it gave the insight to understand the quantitative contributions of local and regional contribution to the VOCs in the study area.

## 3 Results and discussions

### 3.1 VOC levels and compositions

The statistics of 56 observed VOCs are summarized in Table 1 and the hourly variations of four VOC categories are shown in Fig. 3. Among the four different VOC groups, the average concentrations of alkanes were highest (129 ±173 ppbv), followed by alkenes (9.52 ±14.5 ppbv), aromatic hydrocarbons (4.28 ±8.24 ppbv) and acetylene (3.03 ±5.55 ppbv). The top four alkane species were ethane (39.7 ±57.3 ppbv), propane (22.6 ±33.5 ppbv), *n*-butane (15.8 ±21.4 ppbv) and *i*-butane (12.5 ±17.5 ppbv). These four species totally accounted for 64.8% of the alkanes. Among the alkenes, 1-pentene, propylene and ethylene were

the most abundant species with their average concentrations of 4.47 ±6.72 ppbv, 1.88 ±10.2 ppbv and 1.42 ±1.69 ppbv, respectively. They totally represented 71.8% of the alkenes. 96.7% of the aromatic hydrocarbons were composed by benzene, toluene and *m, p*-xylene, with corresponding average concentrations of 1.13 ±1.62 ppbv, 1.06 ±1.91 ppbv and 0.72 ±1.94 ppbv, respectively. Same findings were reported in other oil and natural gas operation and industrial base in the US (Pétron et al., 2012; Helmig et al., 2014; Warneke et al., 2014). For instance, the average concentration of ethane and propane was 74 ±79

15    ppbv and 33d 33ppbv, respectively in winter 2012 collected from Horse pool, Uintah Basin. Despite the highly enhanced VOCs levels was due to the temperature inversion, the VOCs levels in Uintah Basin was still high compared to regional background (Helmig et al., 2014). A distinct chemical signature of collected air samples from the boulder atmospheric observatory in Northeast Colorado was also found to show enhanced concentrations of most alkanes (propane, *n*-butane, *i*-pentane and *n*-pentane) (Pétron et al., 2012). The comparable VOCs levels and compositions between this study and researches

in other oil gas rich areas suggested that VOCs in ambient air was from primary emission from oil and gas development.

   The VOC concentrations, compositions and the top five species in this study and other areas around the world are also compared and shown in Fig. 4. The average VOCs concentrations in this study were 1-50 times higher than those in urban areas like Beijing, Shanghai, Guangzhou, Seoul and 28 cities in the US as well as industrial areas, including Houston, northeast Colorado, Alberta oil sands, Ulsan, YRD and Nanjing (Fig. 4a). As shown in Fig. 3e and Fig. 4b, the alkanes were the most

abundant groups (87.5% on average) during the whole sampling period, which was quite higher than other urban/industrial areas (45.3-67.2%, Fig. 4b). Similar relative high proportions of alkanes were found in Houston (77.1%) (Leuchner and Rappenglück, 2010), Alberta oil sands area (74.8%) (Simpson et al., 2010) and Northeast Colorado (97.4%) (Gilman et al., 2013), which are all related to oil/gas operations. In urban areas, the aromatics accounted for about 10.1-47.9% of the total VOCs, with toluene as one of the most abundant species. Toluene is mainly from solvent usage (Guo et al., 2004; Yuan et al.,

2010) or vehicle exhaust emission (Wang et al., 2010) in cities. Another dominant compound in the urban air is propane (1.45-14.7 ppbv), which is the main component of liquid petroleum gas/natural gas (LPG/NG) (McCarthy et al., 2013). In industrial areas, alkanes and alkenes contribute most to the total VOCs (43.4-97.4% and 1.8-30.0%, respectively) with ethane, propane





and ethylene as the top species usually (Fig. 4c). They may originate from the incomplete combustion or LPG/NG usage (Durana et al., 2006; Tang et al., 2007; Guo et al., 2011). To sum up, the concentrations of VOCs in this study were higher than many other regions and cities. The compositions and the top five species of VOCs exhibited typical characteristics of oil and gas exploring regions, such as Houston (Leuchner and Rappenglück, 2010), North Colorado (Gilman et al., 2013) and

Alberta oil sands area (Simpson et al., 2010).

## 3.2 Contribution of VOCs to ozone formation potential (OFP)

The profiles of different VOC categories with concentrations expressed in different scales are shown in Fig. 5. The top ten VOC species to OFP obtained by Propy-Equiv and MIR method are listed in Table S1. Among the top ten compounds calculated by the two methods, six compounds were the same, but differing in their rank order. Considering the kinetic activity,

1-pentene ranked first by Propy-Equiv method. However, the $o$-xylene showed highest OFP based on the MIR method, which may be related with the chemical mechanisms and the impacts of $NO_x$ (Zou et al., 2015). Despite the two methods were different in mechanisms, the proportions of different VOC categories to the OFP were the same. From the non-weighted concentrations by volume and carbon atom, alkanes contributed $83\pm9\%$ and $82\pm9\%$, respectively to the total VOC concentrations, followed by alkenes ($11\pm6\%$ and $9\pm4\%$, respectively) and aromatics ($5\pm6\%$, and $8\pm7\%$, respectively).

Although the proportions of alkenes and aromatics increased when compared to the values of non-weighted, the alkanes were still dominant, accounting for $45\pm11\%$ and $50\pm14\%$, respectively. In summary, the alkanes had the highest concentrations (for both volume and carbon atom) and largest proportions to the OFP weighted by Propy-Equiv and MIR method. The results of this study were different from previous researches. For example, the alkanes with the highest concentrations (both for volume and carbon atom) contributed less to OFP, while alkenes and aromatics with less concentrations contributed most to the OFP

in Guangzhou (73% and 83%, respectively) (Zou et al., 2015) and Tianjin (about 28-40% and 32-42%) (Liu et al., 2016) as well as a petrochemical industrialized city (48-49% and 37-49%, respectively) (Jia et al., 2016).

## 3.3 Temporal variations

The temporal variation of VOC concentrations is controlled by several factors such as the meteorological parameters, photochemical reactions and emission sources. Due to the lack of information on source emission strengths, we analyzed the

effects of the other two factors on VOC concentrations. The effects of meteorological conditions on VOCs were assessed by Pearson correlation analysis (Fig. S4). The photochemical removal of VOCs was evaluated by analyzing the relationship between the VOCs concentrations and other trace gases (i.e., $NO_2$ and $O_3$) (Fig. 6). Temporal variation of selected parameters and VOCs were analyzed in different timescales.

The daily concentrations of VOCs were found negatively correlated with temperature and wind speed, with $r$ being -0.29

($p<0.01$) and -0.39 ($p<0.01$), respectively. This finding was consistent with previous study (Lyu et al., 2016; Marčiulaitienė et al., 2017). High VOC levels in low temperature are due to more frequent stagnant conditions in cold seasons and proved by the negative correlation between VOCs and BLH ($r$=-0.45, $p<0.01$) in this study. Higher wind speed indicates better dispersion





conditions and thus resulting in lower VOC concentrations. Positive correlations between the VOCs and relative humidity ($r$=0.35, $p$<0.01) and atmospheric pressure ($r$=0.19, $p$<0.01) were found, which was in line with former studies (Marć et al., 2015; Marčiulaitienė et al., 2017).

At 99% confidence interval, the monthly VOC concentrations were positively correlated with $NO_2$ ($r$=0.63, $p$<0.01) (Fig. 6a) and the VOCs were negatively correlated with $O_3$ ($r$=-0.60, $p$<0.01) (Fig. 6b). Similarly, significant correlations were also found between the daily $NO_2$, $O_3$ and VOCs, as shown in Fig. 6c and Fig. 6d. It should be noted that VOCs and $NO_2$ concentrations were controlled by solar UV, concentrations of NO and $O_3$ as well as BLH. For example, the BLH in May, June and July was higher than 900 m, which may explain the lowest $NO_2$ and VOC concentrations during the sampling period. VOC concentrations and trace gases ($NO_2$ and $O_3$) related to photochemical reaction had a pronounced diurnal variation, which is summarized graphically by annual average value in Fig. 7a. Lower BLH and less photochemical activities resulted in the peak values for the precursors of $O_3$ including VOCs and $NO_2$ before sunrise (6:00 local time). After sunrise, with the initiation of photochemical oxidation and the increasing of BLH, the concentrations of $O_3$ precursors decreased and $O_3$ increased rapidly. The minimum of VOCs and $NO_2$ occurred at about 12:00-14:00 LT resulted from both dispersion or dilution conditions and photochemical reactions (with highest $O_3$ concentrations at l4:00 LT) in the afternoon. The similar diurnal patterns of different atmospheric lifetime compounds including ethane, ethylene, acetylene and benzene (the most abundant contributors to its categories) were also found (Fig. S5). To better understand the effects of BLH and photochemical reactions on VOCs, the diurnal variations of VOCs, BLH, and $O_3$ in winter and summer were analyzed (Fig. 7b, c). VOC concentrations in winter (79.9±13.2 to 263±59.3 ppbv) were comparable with that in summer (28.2±9.58 to 261±73.9 ppbv). However, the VOCs in summer and winter decreased by 8.3 times and 2.3 times, respectively, from maximum to minimum. This was due to the BLH increased by 8.2 times in summer while BLH in winter only increased by 2.3 times. The effects of photochemical reactions on VOCs in two seasons were comparable, which was explained by similar $O_3$ increment in winter (0.78 times up) and summer (0.71 times up). Therefore, we can conclude that the role of BLH variation was more important than photochemical reaction for the diurnal variation of VOCs.

Overall, the monthly and daily variations of VOCs were influenced by meteorological conditions and photochemical reactions while the diurnal pattern of VOCs was more influenced by BLH.

### 3.4 Ambient ratios: sources and photochemical removal

Ambient ratios for VOC species holding similar reaction rates with OH radicals can reflect the source features, as these compounds are equally affected by the photochemical processing and the new emission inputs (Russo et al., 2010; Baltrėnas et al., 2011; Miller et al., 2012). For example, *n*-butane and *i*-butane have similar reaction rates with the OH radicals, with the differences less than <10% and the ratios of these pair species indicated different sources. The butanes are associated with NG, LPG, vehicle emission and biomass burning and the ratios of *i*-butane/*n*-butane varied according to sources (i.e., 0.2-0.3 for vehicle, 0.46 for LPG and 0.6-1.0 for NG) (Buzcu and Fraser, 2006; Russo et al., 2010). In this study, the slope of *i*-butane/*n*-



butane (0.80-0.82, Fig. 8a) was within the range of reported emissions from natural gas. Additionally, the slope of *i*-pentane and *n*-pentane was in the range of 1.03-1.24 (Fig. 8b), which was larger than those related to oil and natural gas operations (0.82-0.89) (Gilman et al., 2010, 2013), but less than those measured for liquid gasoline and fuel evaporation (>1.5) (Conner et al., 1995; McGaughey et al., 2004; Watson et al., 2001; Zhang et al., 2013b). The ratios of *i*-pentane/*n*-pentane indicated that the VOCs originated from the mixed sources of NG and fuel evaporation in this study.

Information on the photochemical removal process can be obtained by comparing the ambient ratios of aromatics due to their differences in atmospheric lifetimes. For example, the atmospheric lifetimes of benzene (9.4 days), toluene (1.9 days) and ethylbenzene (1.6 days) are relative longer than *m*-xylene (11.8 h) and *p*-xylene (19.4 h) (Monod et al., 2001). The commonly used ratios are benzene/toluene, *m, p*-xylene/ethylbenzene, benzene/ethylbenzene, and toluene/ethylbenzene. The diurnal variations of these compounds and ratios are shown in Fig. 9. A continuous decreasing of these compounds and ratios were observed from 08:00 to 14:00 LT, indicating the increased photochemical removal processes due to the enhancement of ambient temperature. The diurnal patterns of benzene/ethylbenzene and toluene/ethylbenzene in this study (Fig. 9b, d) was opposite to that observed in Dallas, which was mainly influenced by vehicle emission (Qin et al., 2007). After 14:00 LT, the increasing of the ratios and aromatic concentrations were due to the weakening of photochemical activities. The unusual high concentrations of ethylbenzene and *m, p*-xylene were observed at about 02:00 LT (Fig. 9c), which might be related to new emissions. This assumption was verified by a small peak occurred at 02:00 LT in the diurnal distribution profile of oil refinery source (see section 3.5.1). After 12 h dispersion, dilution and photochemical reaction, the concentrations of these two compounds reached its minimum values at about 14:00 LT.

Generally speaking, when the reaction with OH radicals was the only factor controlling the temporal variation of VOC concentrations, an increase of the concentration ratios for longer atmospheric lifetime over shorter lifetime compounds from lower ambient temperature to higher temperature would be expected (Russo et al., 2010). The diurnal variation of benzene, toluene, ethylbenzene, and xylene (BTEX) obeyed the rule as discussed above, however, the seasonal variation of BTEX ratios was opposite to that exceptionally. For example, the benzene/toluene ratio increased from summer-fall (0.5-0.6) to winter-spring (0.6-0.7) periods (Fig. 8c) and *m, p*-xylene/ethylbenzene ratio decreased from spring-summer (2.5-4.0) to autumn-winter (0.9-1.4) periods (Fig. 8d). The same results were also observed both in industry areas (Miller et al., 2012) and urban areas (Ho et al., 2004; Hoque et al., 2008; Russo et al., 2010). The results obtained in this study indicated that there were other factors affecting the seasonal variation, such as source emissions. BTEX mainly originated from vehicle exhaust (Wang et al., 2010), solvent usage (Guo et al., 2004; Yuan et al., 2010) and petrochemical industry (Na and Kim, 2001; Hsieh et al., 2006; Baltrėnas et al., 2011). As discussed above, the emission from vehicles was not obvious from the diurnal variation of BTEX and the solvent usage was negligible in the study area. The ratios of *m, p*-xylenes/ethylbenzene in this study (2.2±1.2) were within the ranges reported at a petrochemical area in southern Taiwan (1.5-2.6) (Hsieh et al., 2006) and the vicinity of a crude oil refinery at the Baltic region (3.0-4.0) (Baltrėnas et al., 2011). Therefore, BTEX in this area was mainly from the oil refinery emission. The highest *m, p*-xylenes/ethylbenzene ratio in summer can also be related to the highest contribution of oil refinery. This finding was verified by the seasonal source contribution results in section 3.5.1.



### 3.5 Source apportionment: temporal variation and contribution to OFP

Five sources including oil refining process, NG, combustion source, asphalt and fuel evaporation were identified by the PMF analysis and their source profiles and daily contributions are shown in Fig. 10. The monthly, seasonal and annual contributions were calculated and shown in Fig. 11. The relationships between daily source contributions and meteorological parameters and trace gases were analyzed by scatter plots (Fig. 12). The source apportionment of this high-resolution dataset provided a unique opportunity to discuss the diurnal variation of different sources as shown in Fig. 13.

### 3.5.1 Oil refining

The emissions from the refining process are complex due to the diversities of VOCs species, which depend on the production processes (Vega et al., 2011; Mo et al., 2015). The crude oil is composed of $\geqslant$ $C_5$ alkanes, cycloalkanes, aromatics, and asphaltic. These matters are supplied as the raw materials to various oil refining processes (Simpson et al., 2010). High fractions of $C_5$-$C_9$ alkanes including hexane (32±6.2%), cyclohexane (40±7.9%), methylcyclohexane (47±6.9%), n-octane (56±4.2%), n-nonane (58±2.9%), and aromatics (i.e., 22±3.0% for benzene, 39±5.4% for toluene and 45±7.3% for xylenes) presented in this factor (Fig. 10a), which was similar to the chemical structures measured from the oil refinery (Liu et al., 2008b; Dumanoglu et al., 2014). The calculated daily source contributions from PMF model were well correlated with the corresponding species of high loadings in source profile. For example, the cyclohexane showed significant correlation with this source contribution (Fig. S6a), suggesting that the tracers of oil refinery were well produced by the PMF model. The main products from oil refinery are gasoline, diesel, lube, and kerosene in this area, consistent with the factor derived here.

The annual contribution of oil refining source was relatively stable throughout the year (3.8%, 3.2 ppbv). The highest relative contribution was found in summer (5.3%, 2.0 ppbv) and the lowest in winter (2.4%, 2.9 ppbv) (Figure 11b). The Pearson analysis between the daily source contributions and wind speed disclosed a middle statistically negative correlation ($r$=-0.12, $p<0.05$). However, no statistically significant correlations between the daily source contribution and other meteorological parameters were found (Table S2), even for the BLH. On the contrary, statistically significant positive correlations between this source and trace gases ($NO_2$ and CO) were found, with $r$ being 0.33 and 0.21, respectively (Fig. 12a). These trace gases were associated with oil refinery emission (Cetin et al., 2003). Therefore, the daily variation of oil refinery source in this study was more controlled by oil refining emission strength and less influenced by meteorological conditions.

The diurnal pattern of this source contribution was similar with the variation of methylcyclohexane and cyclohexane and characterized by a double wave profile with a first peak at 02:00 LT and second peak at 06:00 LT (Fig. 13a). A small peak occurred at 02:00 was due to the increasing of ethylbenzene and m, p-xylene (Fig. 9) and the second peak occurred at 06:00 LT resulted from the low BLH. After sunrise, the contribution continuously decreased owing to the increasing of BLH and photochemical reactions and the minimum value occurred at 14:00 LT.



### 3.5.2 Natural gas

Ethane and propane are the most abundant non-methane hydrocarbon compounds in the composition of natural gas (Xiao et al., 2008; McCarthy et al., 2013). The ratios of $i$-butane/$n$-butane indicated the butanes were from the natural gas (section 3.3). In the PMF analysis, a NG source was identified through the high weights on ethane ($81 \pm 2.4\%$), propane ($85 \pm 5.3\%$), $n$-butane

($62 \pm 7.5\%$), and $i$-butane ($54 \pm 6.4\%$). As an important oil and gas resources base in China, the export amount of natural gas from this region was $4.4 \times 10^{9}$ $m^3$ and the loss rate was 1.4% in 2014. The leakage from the exploiting, storing, transporting and processing cannot be ignored, suggesting that it was reasonable to attribute this factor to natural gas source.

The annual contribution of the NG leakage source was 53 ppbv, accounting for 62.6% of the total VOCs averagely. The highest contribution presented in spring (65.2%, 38.9 ppbv), followed by summer (63.6%, 43.4 ppbv), autumn (63.0%, 56.7 ppbv),

and winter (60.4 %, 73.1 ppbv). The daily variation of this source was influenced by meteorological parameters such as the BLH ($r$=-0.42, $p$<0.01) (Table S2). The significant positive correlations between $NO_2$ and CO and the source contribution were also found with Pearson coefficients being 0.45 and 0.44, respectively (Fig. 12b), indicating that the daily variation of NG source was influenced by photochemical activities. The diurnal variation of the NG leakage source mostly followed the diurnal pattern of ethane, propane and butanes (Fig. 13b), which was also reported by Baudic et al. (2016). The diurnal

behaviors of this source were characterized by a nighttime high and mid-afternoon low pattern, which can be interpreted as the diurnal pattern of BLH development (Bon et al., 2011; Baudic et al., 2016).

### 3.5.3 Combustion source

The combustion source was dominantly weighted by ethylene ($95 \pm 3.5$ %), acetylene ($97 \pm 2.6\%$) and moderately influenced by BTEX. These species are key markers of combustion (Fujita, 2001; Watson et al., 2001; Jobson, 2004). Known as the

combustion tracer, the CO, $NO_2$, and $PM_{2.5}$ were well correlated with this source contribution, with Pearson coefficients being 0.59, 0.49 and 0.77, respectively (Fig. 12c and Table S2). The source contribution of this factor exhibited obvious seasonal differences with highest contribution in winter (14.9%, 18.1 ppbv) and lowest contribution in summer (6.9%, 4.7 ppbv). This seasonal difference was due to the temperature change and proved by a significant negative correlation with the temperature ($r$=-0.57, $p$<0.01). The diurnal variation of combustion source was in accordance with the diurnal pattern of ethylene, acetylene,

and CO. It was characterized by a double peak profile with an initial increasing from 03:00 to 08:00 LT and a second increasing at nighttime (20:00-0:00 LT) (Fig. 13c). The increase in the morning was related to the low BLH. Different from other researches, no increasing trend of this source was found during 07:00-10:00 LT in this study, while combustion source was reported to be increasing at the rush-hour period (Gaimoz et al., 2011; Baudic et al., 2016). On the contrary, decreasing trends were found for independent combustion tracers (CO and $NO_2$) during this period (Fig. 7a). In another rush-hour of 18:00-

20:00 LT, the enhancement of combustion source contributions and CO from 16:00 LT (Fig. 11c) may be related with the reduction of BLH. The reduction of $NO_2$ from 18:00 LT (Fig. 7a) were also observed, which indicated that the diurnal variation of combustion source was less affected by vehicle exhaust in this study.



### 3.5.4 Asphalt

Asphalt released predominantly $C_8$-$C_{11}$ alkanes including *n*-octane, *n*-nonane, *n*-decane and *n*-undecane, totally contributed to over 50% of VOC emissions from asphalt application (Brown et al., 2007; Liu et al., 2008b; Deygout, 2011), with n-undecane along accounting for 17% (Liu et al., 2008b). Benzene, toluene and xylenes are also enriched for asphalt VOC emissions

(Chong et al., 2014). High loadings of $C_9$-$C_{12}$ VOCs including *n*-nonane, *n*-decane, *n*-undecane, and *n*-dodecane were found in this factor, averaged as 46%, 64%, 72%, and 85%, respectively. The annual processing capacity of heavy oil in this area was $9.0 \times 10^6$ t and the fugitive emission was inevitable.

The annual contribution of this source was the lowest among the five sources and only contributed 1.3% to the total VOCs. The daily contributions of this source and temperature had a statistically reliable positive correlation ($r$=0.19, $p$<0.01). The

seasonal variation of this source was influenced by temperature with the highest contributions occurred in autumn (2.1%) and the lowest in winter (0.5%). However, the influence of BLH on this source contribution was not significant ($r$=0.04, $p$>0.05). The correlations between this source and $O_3$ was found insignificant (r=-0.001, $p$>0.05). However, significant positive correlation between this source and oil refinery source was found ($r$=0.47, $p$<0.01) (Fig. 12d), indicating they shared the same origins (oil refining processes).

The diurnal variation of this source was different from the others. It continuously decreased from 2:00 to 6:00 LT, slowly increased from 6:00 to 10:00 LT and subsequently decreased (Fig. 13d). A minimum source contribution occurred when the BLH was low in the morning, which was contrary to the other sources. An increasing for *n*-dodecane from 06:00 to 18:00 LT was observed, implying less impact of photochemical removal and increasing of the BLH. Therefore, the temporal variation of this source was less controlled by BLH and photochemical reaction, but was more influenced by the emission strength.

### 3.5.5 Fuel evaporation

The gasoline evaporation profile holds high proportions of *i*-pentane, *trans*-2-pentene, *cis*-2-pentene, benzene, and toluene (Liu et al., 2008b; Zhang et al., 2013b). The *i*-pentane is a key tracer of gasoline evaporation due to its high abundance (Gentner et al., 2009; Zhang et al., 2013b). The ratios of *i*-pentane to *n*-pentane is useful to identify the potential sources including NG (0.82-0.89), liquid gasoline (1.5-3.0), fuel evaporation (1.8-4.6), and vehicle emission (2.2-3.8) (Harley et al., 2001;

McGaughey et al., 2004; Russo et al., 2010; Gilman et al., 2013). As discussed above, the ratios of *i*-pentane/*n*-pentane indicated a mixed source in this region. From PMF results, high loadings on *i*-pentane and *n*-pentane were present in this factor profile, which accounted for 85±5.3% and 71±6.4% of total species, respectively. Additionally, this factor was influenced by hexane (60±5.6%), cyclohexane (45±10%), methylcyclohexane (52±7.5%), benzene (23±3.0%), and toluene (19±4.0%), which were related to diesel fuel evaporation (Liu et al., 2008b). As shown in Fig. S1, the products of oil refinery included the

gasoline and diesel, with the annual productions of $9.5 \times 10^5$ t and $1.9 \times 10^5$ t, respectively. Therefore, this factor represented the fuel evaporation.



The fuel evaporation is fugitive and controlled by temperature, leading to higher contributions in summer. The highest contribution was found in summer as 22.9% in this study. The same results were also observed previously (Baudic et al., 2016; Liu et al., 2016). A significant correlation between the contributions of NG and fuel evaporation was found ($r$=0.65, $p$<0.01), indicating the contributions of two types of sources were influenced by similar factors. The diurnal distribution pattern of fuel evaporation source was different from former studies in urban area (with an increasing trend from 7:00 to 10:00 LT due to the morning rush hour) (Baudic et al., 2016).

### 3.5.6 Contribution to OFP

The contributions of five identified VOC sources to OFP were also evaluated using $F$ matrix and MIR method (Eq. 4). The fuel evaporation showed highest contribution (41.9%, 41.6 ppbv), followed by NG (29.6%, 29.4 ppbv), combustion (14.2%, 14.1 ppbv), oil refinery (11.3%, 11.2 ppbv) and asphalt (3.0%, 3.0 ppbv). Therefore, more attention should be paid to fuel evaporation due to its high ozone formation potential. It should be noted that the source contributions to OFP were calculated by 20 selected VOC species in PMF model analysis and the actual contributions to OFP were higher than the results.

### 3.6 Source contributions compared with previous studies

The source apportionment results showed that the dominant source in this study was the natural gas source, contributing 63% to the total VOCs on the annual average, followed by fuel evaporation (22%), combustion source (11%), oil refinery (3.8%), and asphalt emission (1.3%). Each identified PMF factor exhibited obvious temporal variations due to emission strength, photochemical reaction and meteorological parameters. The source apportionment results in this study were compared with formers based on long time series monitoring (Table 2).

The contributors to VOCs in urban area were complex with at least five different sources, including fuel evaporation, LPG/NG, industrial emission, vehicle emission, and solvent usage (Table 2). While the number of VOC sources apportioned in industrial areas was less compared to the cities. For example, only three sources including vehicle emission (58.3%), solvent usage (22.2%), and industrial activities (19.5%) were apportioned by principle component analysis-multiple linear regression (PCA-MLR) in Lanzhou, a petrochemical industrialized city in northwest China (Jia et al., 2016). Same results were also found in Houston that only fuel evaporation, industrial emission, and vehicle emission were identified (Leuchner and Rappenglück, 2010). In these studies, the vehicle emission was an important source both in urban and industrial areas and contributed about 11-58.3% to the total VOCs (Table 2). However, the vehicle emission source was not identified in this study due to several reasons. Firstly, despite there was similarity between the source profiles of combustion/fuel evaporation in this study and the vehicle emission (e.g., high loadings on acetylene, ethylene, BTEX, butanes, and pentanes), the temporal variation of these species did not show a distinct increasing at the traffic rush-hour. In fact, the identified combustion source in this study represented the characteristics of coal burning and torch burning in oil refinery (to eliminate the hazardous gases). Secondly, differences existed in sampling location and vehicle amounts. In previous urban studies, the sampling location was in megacities with huge vehicle flow. For example, in the research of Wuhan (Lyu et al., 2016), the sampling site located in the



city center and the car ownership was $2.2 \times 10^6$ by the end of 2015. While the sampling location in this study was about 11 km away from the urban areas and the car ownership was only $1.1 \times 10^5$. Therefore, the factor with higher loadings of these species was not likely to be contributed by vehicle emission in this study.

LPG and NG sources are usually apportioned both in urban and industrial areas. These sources contribute 10%-32% to the total VOCs and are mainly from household or industrial fugitive emission. However, in this study, the NG source was mainly from the NG exploitation and NG chemical industry due to its abundance in this area and accounted for 62.6% on average to VOCs. Which was higher than many other areas as summarized in Table 2.

Solvent usage also accounts for large proportion of total VOCs in urban areas (4.7-36.4%). In this study, a similar source related to asphalt was identified with heavy weights on $C_9 \sim C_{12}$ compounds. The solvent usage in urban areas is usually from painting/coating. However, the asphalt in this study originated from oil refinery (Fig. S1) and fugitive emission from a black oil hill located in the northwest of the sampling site. Due to its high boiling point, the seasonal contribution of asphalt was distinct with highest contribution in July (7.2%) and lowest contribution in January (1.4%). Despite the source contribution of asphalt was low, it was unique in this study.

### 3.7 Geographic origins of VOC sources: local vs. regional contributions

The possible geographic origins of five identified VOC sources were explored by CPF, PSCF and CWT as shown in Fig. 14, Fig. 15 and Fig. 16, respectively. These methods aimed at providing insights on the potential geographic origins of VOCs sources but did not claim to be precise at the cell level or pixel level.

Highest CPF values of oil refinery was found in the east direction (Fig. 14a, 14b and 14d), indicating the location of this source. In fact, the oil refinery is mainly located in the southwest of sampling site (Fig. 1c) and high CPF value (0.95) was also found in southwest direction Therefore, the CPF results located the oil refineries well. Similarly, high probabilities and concentrations of oil refinery were found from northeast to southwest area of the sampling site according to the PSCF (Fig. 15a) and CWT plots (Fig. 16a). As shown in Fig. 1a and 1b, the sampling site is located in the west of the Junggar Basin, which is the second largest oil-gas basin of China. However, the CPF and PSCF analysis of NG source did not exhibit high probabilities from east direction of the sampling site (Fig. 14b, Fig. 15b), while high CWT values of this source occurred in east direction (Fig. 16b). Given the fact that NG source was composed by long atmospheric lifetime species (i.e., ethane, propane and butanes), the high probabilities and concentrations of this factor are likely resulted from aged air masses from each direction. The combustion source showed high potential from ESE to SE directions according to the CPF, PSCF, and CWT plots. There were no high values in the northwest direction of sampling site, where the urban area locates. This also indicated that the combustion from vehicle emission was insignificant in this study. For asphalt source, highest CPF value was found in the east direction while the PSCF and CWT plots showed high values in the northeast direction. As discussed above, the asphalt source in this study were from the natural source (black oil hill in the northwest of sampling site) and oil refinery (southwest direction). The CPF, PSCF, and CWT results indicated that these methods failed to locate the natural source of asphalt. The potential geographic origins of fuel evaporation were widespread from ESE to W directions, which was similar to the oil refinery source.




Differences of geographic origins were also found in different seasons (Fig. S7-S14). The potential source areas of the five sources spread from northeast to southwest in autumn. In winter, both PSCF and CWT methods indicated that the VOC sources were probably from the southeast and southwest. In spring, VOCs were mainly from long-range transport from west. However, high probabilities and contributions existed around the sampling site. In summer, high potentials and contributions were from the west to the southeast direction. Overall, the five sources exhibited different local source areas proved by the CPF plots on the annual scale. Similar regional distributions of these sources were found on the seasonal scale. To quantify the contributions of local emission and long-range transport to the sampling site, raster analysis based on CWT was used and the results were summarized in Table 3. Annually, except for the combustion source, the identified VOCs sources were mainly from the local emission, with contributions of 53.6% for oil refining, 54.5% for NG, 50.5% for asphalt and 50.6% for fuel evaporation, respectively. The seasonal patterns were same with the annual pattern, exhibiting higher local contributions and differences only existed in the proportions. The highest local contributions of oil refining (69.4%) and combustion (69.2%) were observed in summer, while the local sources contributed most to NG (74.6%), asphalt (65.4%) and fuel evaporation (68.3%) in autumn.

## 4 Conclusions

Based on one-year continuously online monitoring of VOCs in an oil gas field, and on the use of PMF receptor model, back trajectory, PSCF, and CWT dispersion models, this study allowed for (ⅰ) the comparison of VOC levels and compositions with other studies, (ⅱ) the temporal variation of the total VOCs at different time scale (seasonal, monthly and diurnal), (ⅲ) the source apportionment of VOCs, and (ⅳ) the exploration of the potential geographic origins of five identified VOCs sources. The main findings are summarized as follows:

1. The VOC concentrations in this study were not only higher than those in urban areas, but also higher than those measured in petrochemical areas. The VOC compositions in this study were similar to those observed in petrochemical areas such as Uintah Basin and Northeast Colorado, with the alkanes contributed most (87.5%, 128 ±82.4 ppbv) to the total VOCs, followed by alkenes (6.81%, 9.1±5.6 ppbv), aromatic hydrocarbons (3.37%, 4.8±6.5 ppbv), and acetylene (2.32%, 3.1±5.1ppbv).

2. The monthly and daily variations of the total VOCs were influenced by meteorological conditions and photochemical reactions while the diurnal pattern of VOCs was more impacted by boundary layer height.

3. Five sources with local characteristics were identified. The NG contributed most to the VOCs (62.6%), followed by fuel evaporation (21.5%), combustion source (10.9%), oil refining (3.8%), and asphalt (1.3%). The NG and fuel evaporation source contributions showed positive correlation with each other and shared the same diurnal pattern, exhibiting a single peak profile. The diurnal pattern of oil refining and combustion source exhibited similar double wave with peaks occurred 06:00-08:00 LT. Different from other sources, the diurnal profile of asphalt exhibited a decreasing trend from nighttime to its minimum before sunrise (06:00 LT).

4. The geographic origins of five VOC sources were the same during the whole period. The differences existed in the seasonal variations of them. For instance, VOCs were mainly from northeast and southwest in autumn, while it originated from southeast





and southwest in winter. The raster analysis indicated that the VOCs in this study were mainly from local emission with contribution ranged from 48.4% to 74.6% in different seasons.

In summary, this study found that the VOC concentrations, compositions, ozone formation potential, and sources were different from other areas and similar to oil-gas rich areas. It will be helpful for the VOCs control in these type of regions
5   around the world.

**Data availability**

The VOC concentrations during the whole sampling period are available on request from Shaofei Kong (kongshaofei@cug.edu.cn).

*Acknowledgements.* This study was financially supported by the Key Program of Ministry of Science and Technology of the
10   People's Republic of China (2016YFA0602002; 2017YFC0212602). The authors are grateful to the local Environmental Monitoring Center Station for their works in sampling campaign. We thank Qingyue Open Environmental Data Center (https://data.epmap.org) for providing air quality data.





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



**Table 1.** Concentrations (mean ± standard deviation) and photochemical properties of VOCs during the sampling period.

| Species | $r^2$ | MDL | MIR [a] | KOH ($\times 10^{12}$) [b] | ppbv | ppbC |
|---|---|---|---|---|---|---|
| Alkanes | | | | | 129±173 | 387±439 |
| **Ethane** [c] | 0.997 | 0.05 | 0.25 | 0.25 | 39.7±57.3 | 53.2±76.7 |
| **Propane** | 0.999 | 0.021 | 0.48 | 1.09 | 22.6±33.5 | 44.5±65.7 |
| *i*-**Butane** | 0.994 | 0.012 | 1.21 | 2.12 | 12.5±17.5 | 32.3±45.2 |
| *n*-**Butane** | 0.994 | 0.03 | 1.02 | 2.36 | 15.8±21.4 | 40.8±55.5 |
| Cyclopentane | 0.997 | 0.026 | 2.4 | 4.97 | 8.64±16.0 | 26.7±50.1 |
| *i*-**Pentane** | 0.998 | 0.012 | 1.38 | 3.6 | 8.96±13.3 | 28.8±42.7 |
| *n*-**Pentane** | 0.984 | 0.026 | 1.04 | 3.94 | 8.81±12.4 | 28.3±39.7 |
| 2,2-Dimethylbutane | 0.998 | 0.007 | 0.82 | 2.23 | 0.24±0.60 | 0.92±2.30 |
| 2,3-Dimethylbutane | 0.999 | 0.005 | 1.07 | 5.78 | 1.81±2.91 | 6.94±11.2 |
| 2-Methylpentane | 0.984 | 0.005 | 1.5 | 5.2 | 3.36±5.96 | 12.9±22.9 |
| 3-Methylpentane | 0.998 | 0.007 | 1.5 | 5.2 | 1.40±2.77 | 5.38±10.6 |
| *n*-**Hexane** | 0.997 | 0.016 | 0.98 | 5.2 | 3.08±4.88 | 11.8±18.7 |
| 2,4-Dimethylpentane | 0.999 | 0.005 | 1.5 | 4.77 | 0.13±0.56 | 0.56±2.49 |
| Methylcyclopentane | 0.999 | 0.008 | 2.8 | - | 1.28±3.30 | 4.81±12.4 |
| 2-Methylhexane | 0.996 | 0.008 | 1.08 | - | 0.57±1.19 | 2.56±5.31 |
| **Cyclohexane** | 0.997 | 0.004 | 1.28 | 6.97 | 1.23±2.03 | 4.61±7.61 |
| 2,3-Dimethylpentane | 0.997 | 0.016 | 1.31 | - | 0.85±1.37 | 3.79±6.11 |
| 3-Methylhexane | 0.995 | 0.006 | 1.4 | - | 0.77±1.39 | 3.42±6.21 |
| 2,2,4-Trimethylpentane | 0.994 | 0.003 | 0.93 | 3.34 | 0.05±0.42 | 0.23±2.14 |
| n-Heptane | 0.994 | 0.007 | 0.81 | 6.76 | 4.07±30.5 | 18.2±136 |
| **Methylcyclohexane** | 0.995 | 0.008 | 1.8 | 9.64 | 1.43±2.47 | 7.28±12.6 |
| 2,3,4-Trimethylpentane | 0.994 | 0.008 | 1.6 | 6.6 | 0.06±0.48 | 0.32±2.42 |
| 2-Methylheptane | 0.99 | 0.008 | 0.96 | - | 0.87±1.41 | 4.43±7.20 |
| 3-Methylheptane | 0.991 | 0.009 | 0.99 | - | 0.20±0.53 | 1.03±2.71 |
| *n*-**Octane** | 0.989 | 0.121 | 0.6 | 8.11 | 0.97±1.28 | 4.93±6.51 |
| *n*-**Nonane** | 0.998 | 0.021 | 0.54 | 9.7 | 0.29±0.59 | 1.68±3.37 |
| *n*-**Decane** | 0.995 | 0.03 | 0.46 | 11 | 0.23±0.32 | 1.44±2.05 |
| *n*-**Undecane** | 0.992 | 0.02 | 0.42 | 12.3 | 0.19±0.24 | 1.29±1.66 |
| *n*-**Dodecane** | 0.993 | 0.01 | 0.38 | 13.2 | 39.7±57.3 | 53.2±76.7 |
| Alkenes | | | | | 9.52±14.5 | 30.6±41.7 |
| **Ethylene** | 0.997 | 0.003 | 7.4 | 8.52 | 1.42±1.69 | 1.78±2.11 |
| Propylene | 0.998 | 0.025 | 9.4 | 26.3 | 1.88±10.2 | 3.53±19.1 |
| *trans*-2-butene | 0.997 | 0.031 | 10 | 64 | 0.60±1.34 | 1.50±3.34 |
| 1-Butene | 0.994 | 0.03 | 8.9 | 31.4 | 0.63±1.04 | 1.58±2.59 |
| *cis*-2-butene | 0.999 | 0.023 | 10 | 56.4 | 0.70±2.06 | 1.75±5.15 |
| 1-Pentene | 0.993 | 0.03 | 6.2 | 31.4 | 4.47±6.72 | 14.0±21.0 |
| *trans*-2-Pentene | 0.998 | 0.009 | 8.8 | 67 | 0.19±0.67 | 0.60±2.11 |
| Isoprene | 0.998 | 0.008 | 9.1 | 101 | 0.20±0.75 | 0.62±2.28 |
| *cis*-2-Pentene | 0.998 | 0.015 | 8.8 | 65 | 0.09±0.25 | 0.28±0.80 |
| 1-Hexene | 0.984 | 0.008 | 4.4 | 37 | 1.36±2.74 | 5.09±10.3 |
| **Acetylene** | 0.998 | 0.048 | 0.5 | | 3.03±5.55 | 3.52±6.44 |
| Aromatics | | | | | 4.28±8.24 | 22.2±25.3 |
| **Benzene** | 0.997 | 0.007 | 0.42 | 1.22 | 1.13±1.62 | 3.95±5.66 |
| **Toluene** | 0.995 | 0.005 | 2.7 | 5.63 | 1.06±1.91 | 4.34±7.84 |
| Ethylbenzene | 0.992 | 0.003 | 2.7 | 7 | 0.30±2.40 | 1.41±11.4 |





| | | | | | | |
|---|---|---|---|---|---|---|
| *m*,*p*-Xylene | 0.986 | 0.002 | 7.4 | 18.7 | 0.72±1.94 | 3.42±9.19 |
| *o*-Xylene | 0.989 | 0.003 | 6.5 | 13.6 | 0.20±0.59 | 0.95±2.79 |
| Styrene | 0.991 | 0.013 | 2.2 | 58 | 0.40±2.60 | 1.87±12.1 |
| *iso*-Propylbenzene | 0.986 | 0.02 | 2.2 | 6.3 | 0.06±0.24 | 0.32±1.28 |
| *n*-Propylbenzene | 0.986 | 0.016 | 2.1 | 5.8 | 0.05±0.16 | 0.29±0.85 |
| *m*-ethyltoluene | 0.989 | 0.02 | - | 18.6 | 0.09±0.20 | 0.49±1.06 |
| *p*-ethyltoluene | 0.992 | 0.02 | - | 11.8 | 0.08±0.18 | 0.44±0.98 |
| 1,3,5-Trimethylbenzene | 0.989 | 0.004 | 10.1 | 56.7 | 0.09±0.18 | 0.47±0.96 |
| *o*-ethyltoluene | 0.999 | 0.02 | - | 11.9 | 0.07±0.18 | 0.35±0.95 |
| 1,2,4-Trimethylbenzene | 0.991 | 0.003 | 8.8 | 32.5 | 0.14±0.25 | 0.74±1.34 |
| 1,2,3-Trimethylbenzene | 0.993 | 0.002 | 8.9 | 32.7 | 0.09±0.16 | 0.46±0.85 |
| *m*-diethylbenzene | 0.991 | 0.02 | - | - | 0.07±0.08 | 0.40±0.50 |
| *p*-diethylbenzene | 0.993 | 0.03 | - | - | 0.10±0.10 | 0.57±0.61 |

[a] Units: g $O_3$/g VOCs (Carter, 1994).
[b] Units: ×$10^{-12}$ $cm^3$ molecule$^{-1}$ s$^{-1}$ (Atkinson and Arey, 2003)
[c] Species in bold were used in source apportionment



**Table 2** Comparison of VOCs source apportionment results with formers

| Area | Sampling period | Model | Sources | | | | | | | | |
|---|---|---|---|---|---|---|---|---|---|---|---|
| | | | Fuel evaporation | LPG/NG | Industrial emission | Vehicle emission | Solvent usage | Coal or biomass | Stationery +mobile | Biogenic | others |
| Tianjing, urban [a] | Nov 2014-Oct 2015 | PMF | 8.7 | 18.6 | 19.9 | 39.1 | 4.7 | 10.6 | | | 9.0 |
| Wuhan, urban [b] | Feb 2013-Oct 2014 | PMF | | 19.8±0.9 | 14.4±0.9 | 27.8±0.9 | 16.2±0.4 | 21.8±0.9 | | | |
| Lanzhou, downtown [c] | Jan-Dec 2013 | PCA-MLR | | | 19.5 | 58.3 | 22.2 | | | | |
| Paris, urban [d] | Jan 2010-Dec 2010 | PMF | 5 | 16 | | 23 | 26 | 13 | | 17 | |
| Nanjing, industrial area [e] | Mar 2011-Feb 2012 | PCA/APCS | | 15–48 | 15–23 | 29–50 | 6–15 | | | 1-4 | 15-23 |
| Hong Kong, urban [f] | Jan 2001-Dec 2001 | PCA/APCS | | 11–19.4 | 5.2–9 | 38.9–48 | 32–36.4 | | | 0.1 | |
| Paterson, urban [g] | Nov 2005-Dec 2006 | PMF | | | 16 | 31 | 19 | | 12 | | 22 |
| Houston,industrial area [h] | Aug 2006-Sep 2006 | PMF | | 20–37 | 39–58 | 11–16 | | | | | |
| Los Angeles, urban [i] | 2001-2003(Jul-Sep) | PMF | 47-58 | 13 | 15 | 22–24 | | | | 1-3 | |
| This study | Sep 2014-Aug 2015 | PMF | 21.5 | 62.6 | 3.8 | | | 10.9 | | | 1.3 |

[a] (Liu et al., 2016)
[b] (Lyu et al., 2016)
[c] (Jia et al., 2016)
[d] (Baudic et al., 2016)
[e] (An et al., 2014)
[f] (Guo et al., 2007)
[g] (Yu et al., 2014)
[h] (Leuchner and Rappenglück, 2010)
[i] (Brown et al., 2007)



**Table 3** Contributions (%) of local sources and regional transport of five sources in different seasons.

| Seasons | Oil refining | | NG | | Combustion source | | Asphalt | | Fuel evaporation | |
|---------|------|----------|------|----------|------|----------|------|----------|------|----------|
| | Local | Regional | Local | Regional | Local | Regional | Local | Regional | Local | Regional |
| Autumn | 64.5 | 35.5 | 74.6 | 25.4 | 68.6 | 31.4 | 65.4 | 34.6 | 68.3 | 31.7 |
| Winter | 60.1 | 39.9 | 60.0 | 40.0 | 58.5 | 41.5 | 60.3 | 39.7 | 59.0 | 41.0 |
| Spring | 66.5 | 33.5 | 66.0 | 34.0 | 59.7 | 40.3 | 64.0 | 36.0 | 60.0 | 40.0 |
| Summer | 69.4 | 30.6 | 71.9 | 28.1 | 69.2 | 30.8 | 62.7 | 37.3 | 65.9 | 34.1 |
| Annual | 53.6 | 46.4 | 54.5 | 45.5 | 48.8 | 51.2 | 50.5 | 49.5 | 50.6 | 49.4 |




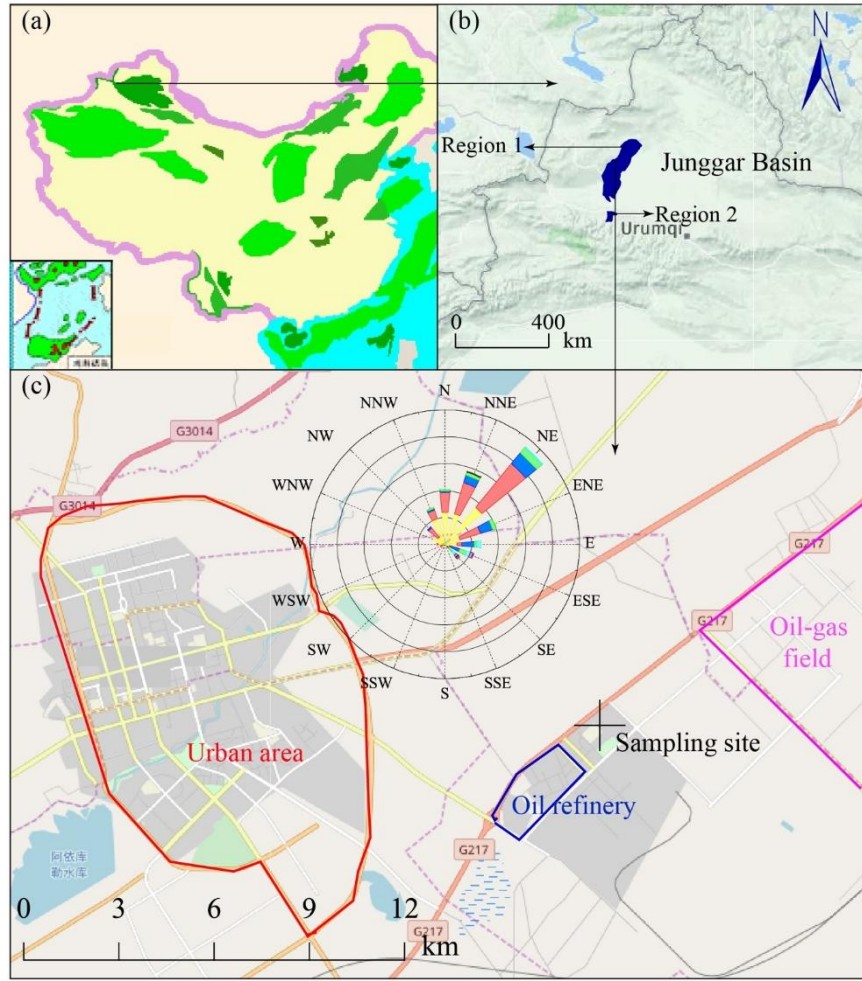

**Figure 1.** The spatial distribution of oil gas bearing basins in China (a) and the terrain of the study area (b). The sampling site is about 11 km away from the urban area and located in the northeast of an oil refinery plant and southwest of an oil-gas field (c). Northwesterly and northeasterly winds prevailed during the sampling periods.





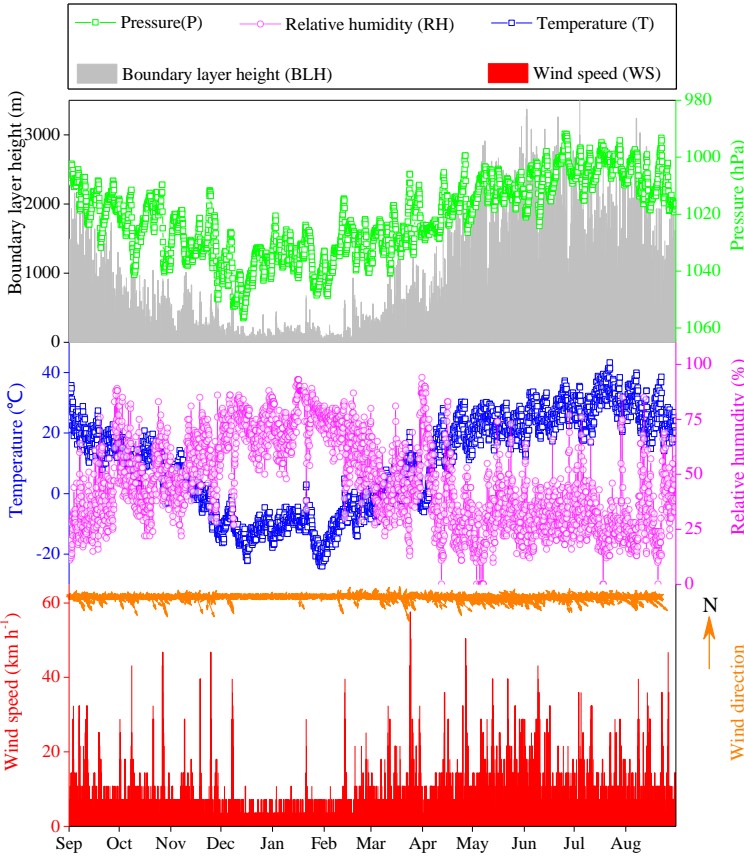

**Figure 2.** Hourly variations of meteorological parameters at the observation site from September 2014 to August 2015.





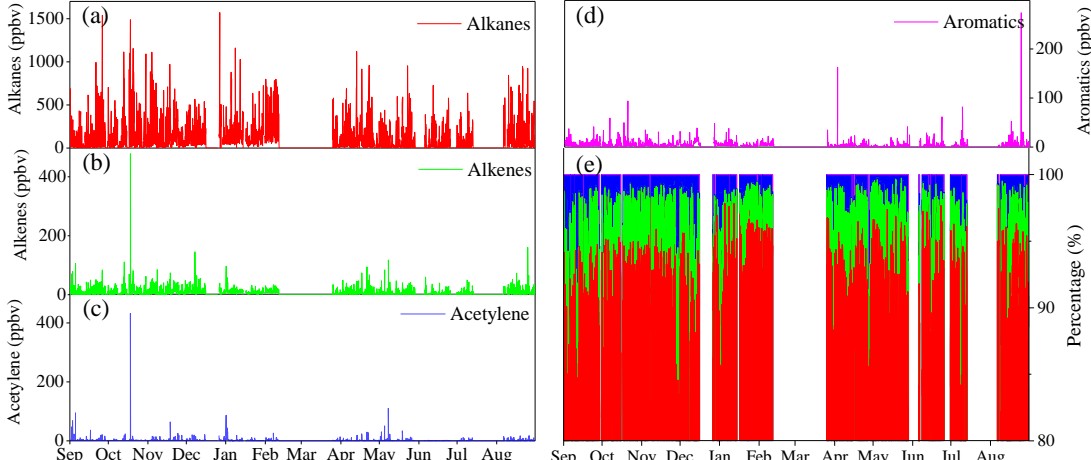

**Figure 3**. Time series of the hourly concentrations (expressed in ppbv) of four categories of VOCs including alkanes (a), alkenes (b), acetylene (c), aromatics (d), and their fractions (e) during the sampling period.



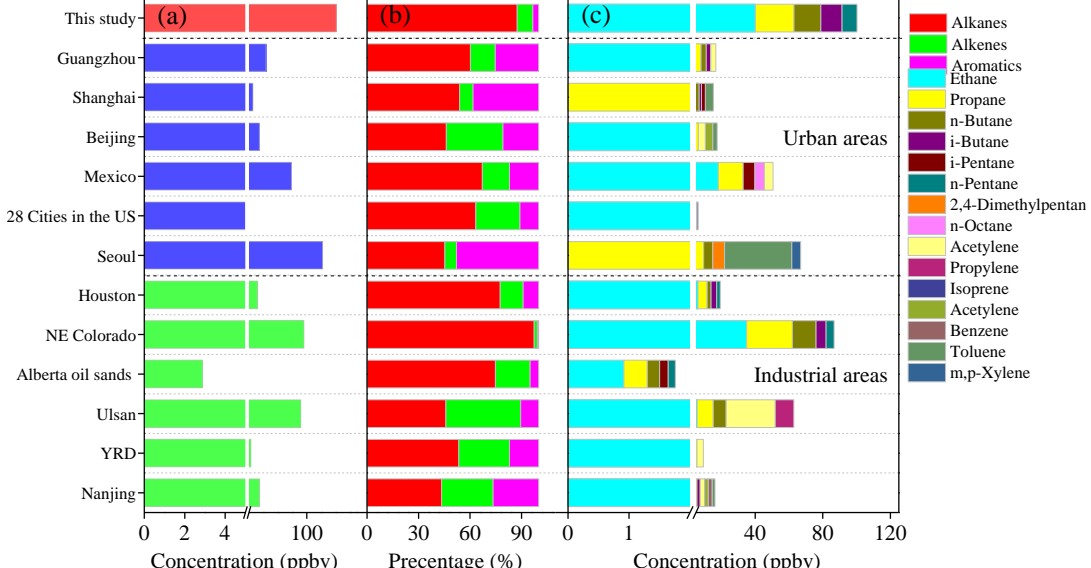

**Figure 4.** Comparison of the VOC concentrations (a), compositions (b) and the top five VOC species (c) in this study and former studies concerning the VOCs in the ambient of urban and industrial areas.



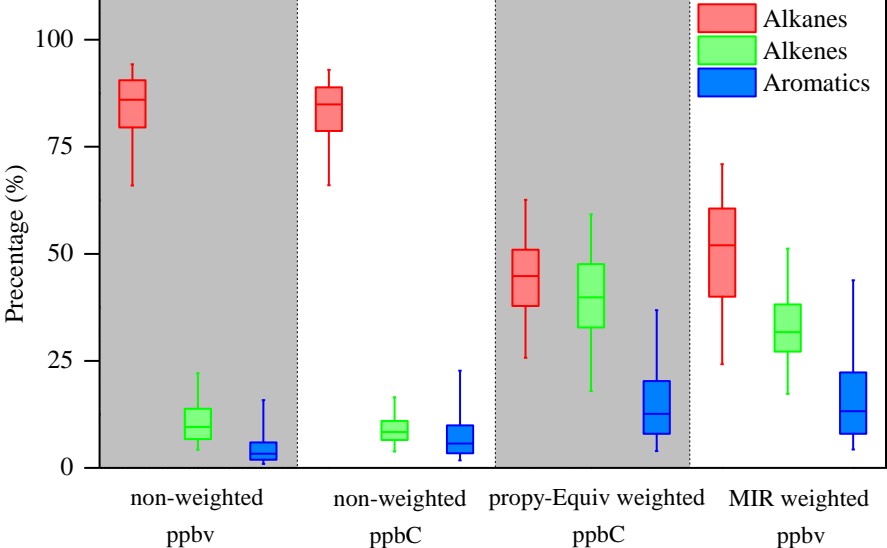

**Figure 5.** Box and whisker plots of VOC profiles based on different scales during the whole sampling period. Box and Whisker plots are constructed according to 25th-75th and 5th -95th percentile of the calculation results.





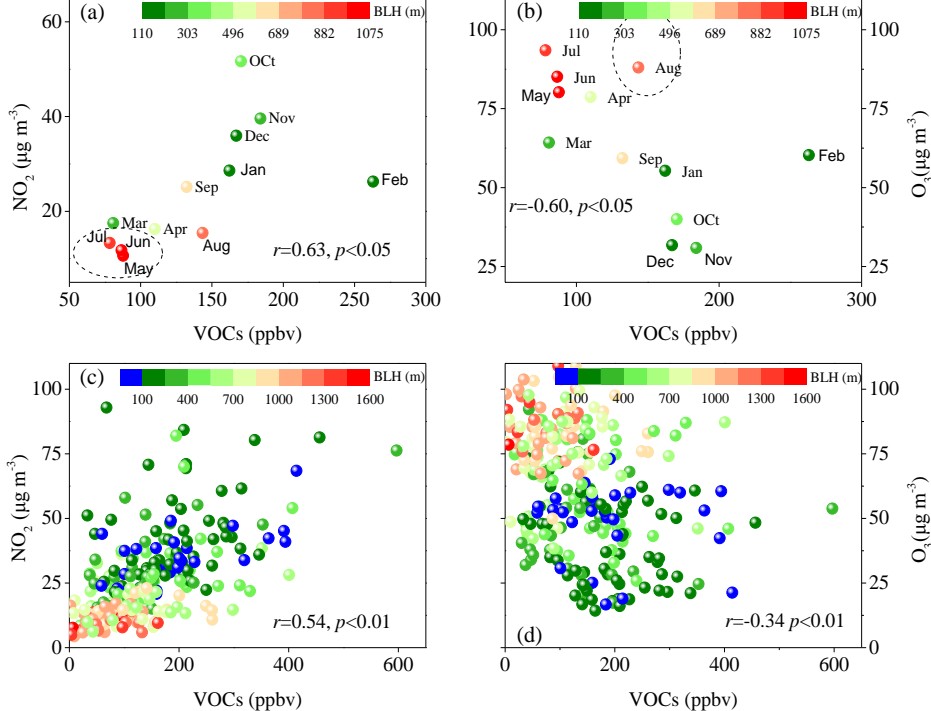

**Figure 6.** Scatter plots of VOCs, NO$_2$ and O$_3$ under different boundary layer height (BLH) at monthly (a and b) and daily timescale (c and d).



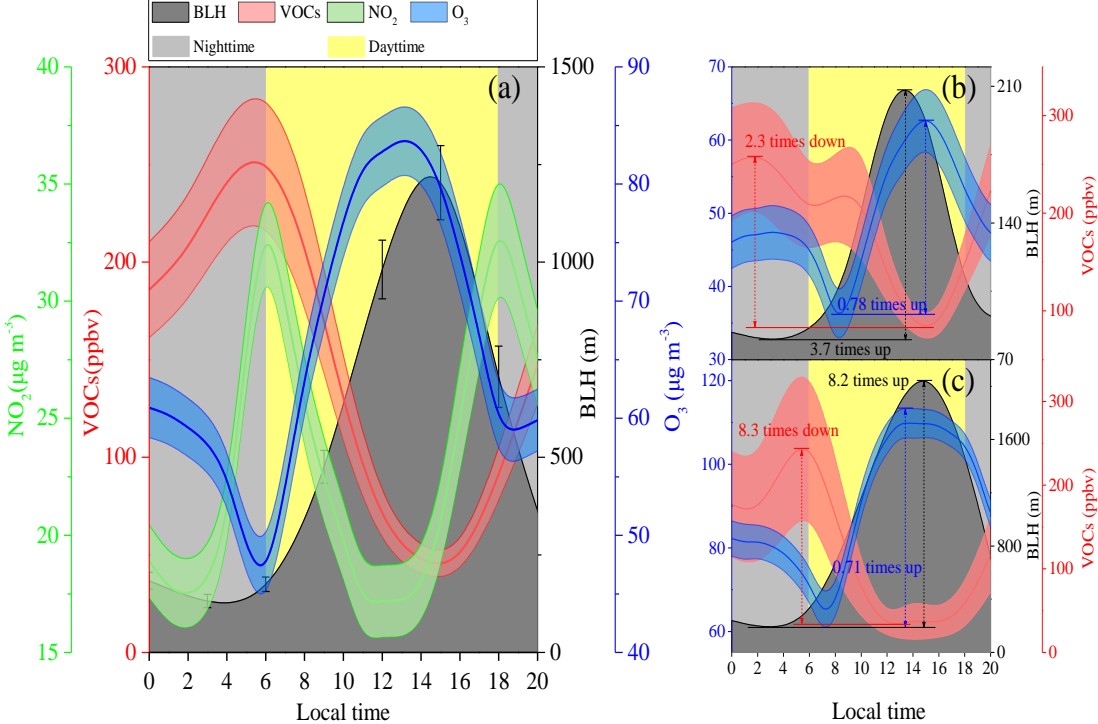

**Figure 7.** Diurnal variation of boundary layer height (BLH), VOCs, $NO_2$, and $O_3$ concentrations in different timescale: annual (a), winter (b) and summer (c). Solid line represents the average value and filled area indicates the 95th confidence intervals of the mean.





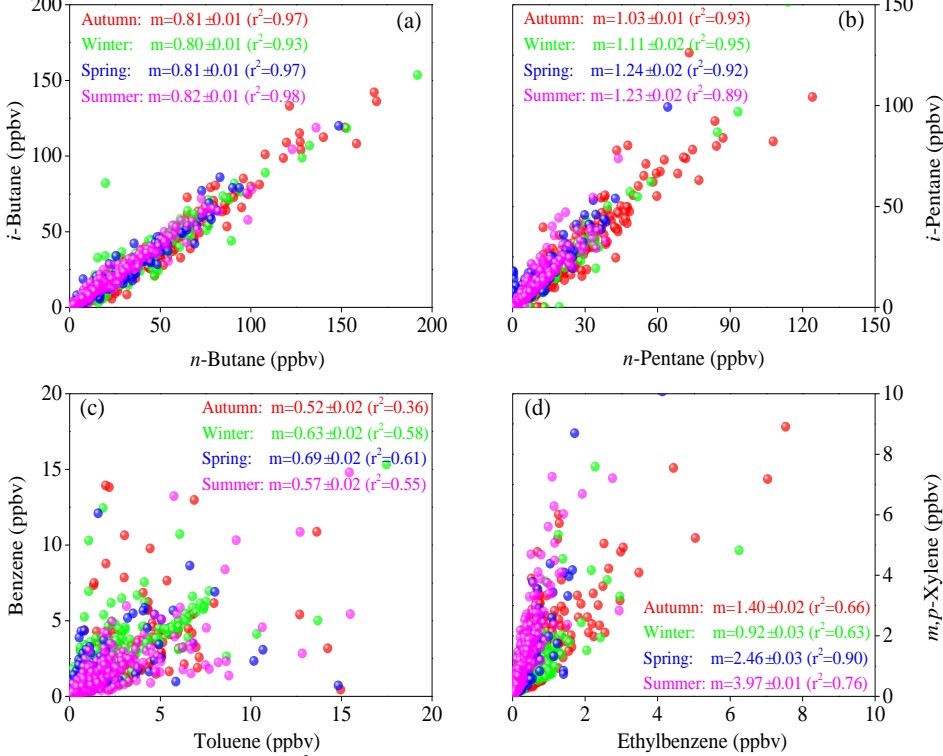

**Figure 8.** Correlations (m=slope ± standard error (r²)) between compounds with similar atmospheric lifetimes including *n*-butane*/i*-butane (a) and *n*-pentane*/i*-pentane (b), and compounds with different lifetimes including benzene/toluene (c) and *m, p*-xylenes*/*ethylbenzene (d).





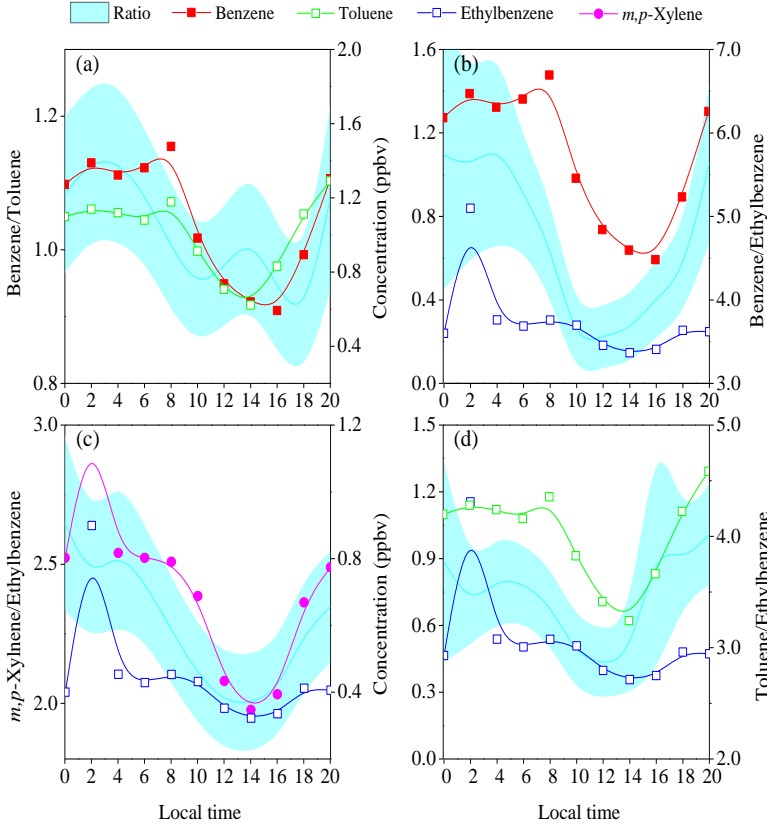

**Figure 9.** Diurnal variations of benzene, toluene, ethylbenzene, and *m, p*-xylene and their ratios: benzene/toluene (a), benzene/ethylbenzene (b), *m, p*-xylene/ethylbenzene (c), and toluene/ethylbenzene (d).





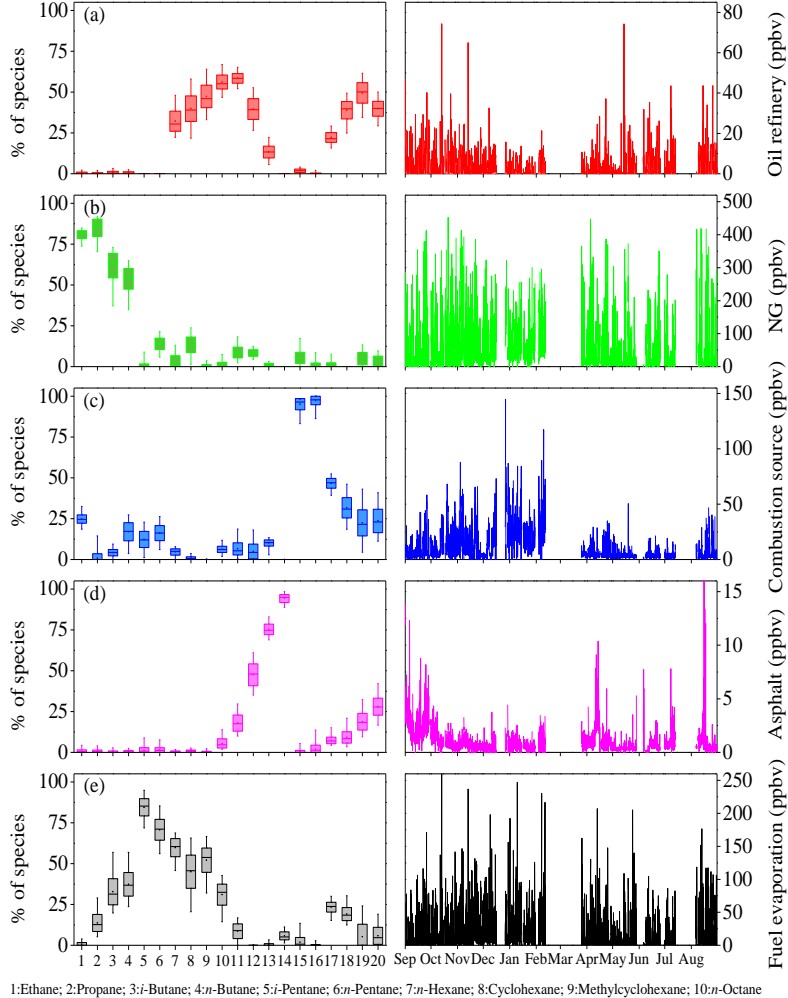

1:Ethane; 2:Propane; 3:*i*-Butane; 4:*n*-Butane; 5:*i*-Pentane; 6:*n*-Pentane; 7:*n*-Hexane; 8:Cyclohexane; 9:Methylcyclohexane; 10:*n*-Octane
11:*n*-Nonane; 12:*n*-Decane; 13:*n*-Undecane; 14:*n*-Dodecane; 15:Ethylene; 16:Acetylene; 17:Benzene; 18:Toluene; 19:*m,p*-Xylene; 20:*o*-Xylene

**Figure 10.** Source profiles of five factors resolved by PMF including oil refinery (a), NG (b), combustion source (c), asphalt (d) and fuel evaporation (e), and their corresponding hourly source contributions. Box and whisker plots are constructed according to the 5th-95th percentiles of the F-peak bootstrap runs (n=100).





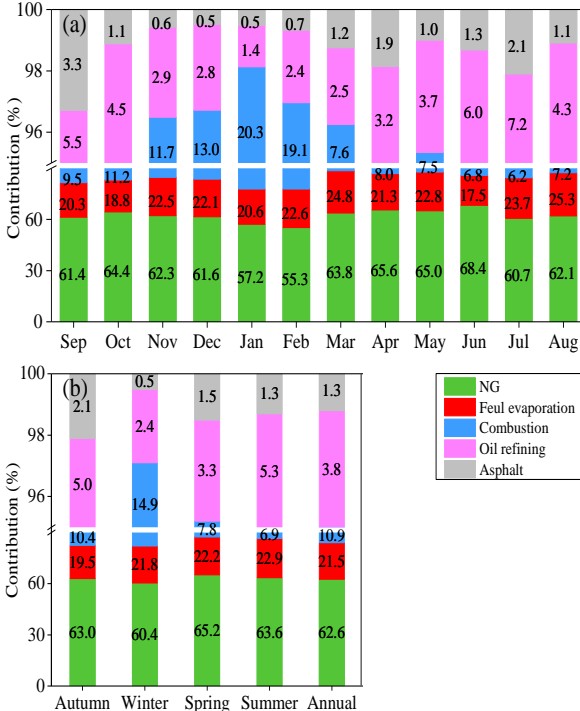

**Figure 11.** Variations of monthly averaged (a) and seasonal averaged (b) contributions of five identified VOC sources (expressed in %).



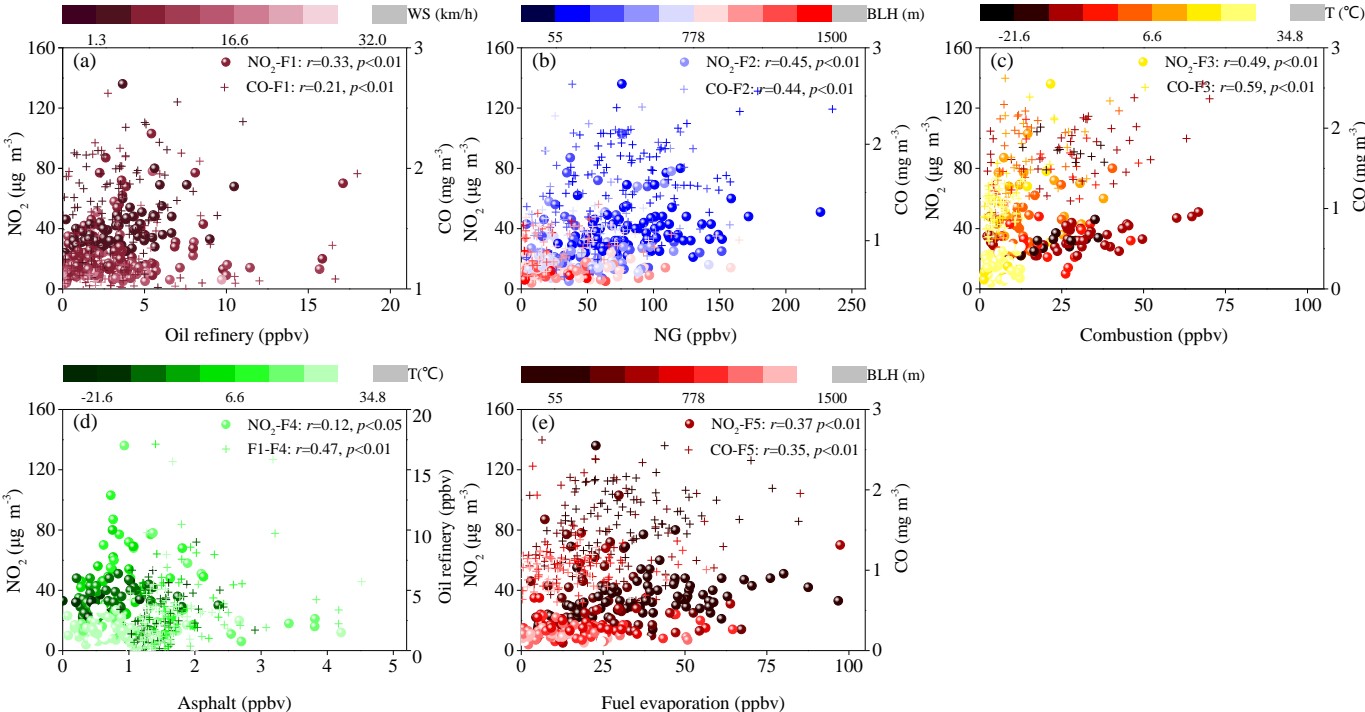

**Figure 12.** Scatter plots of daily concentrations of trace gas and source contributions including oil refinery (a), NG (b), combustion (c), asphalt (d), and fuel evaporation (e) under different meteorological conditions (wind speed (WS), boundary layer height (BLH) and temperature (T)).





**Figure 13.** Diurnal variations of the contributions (expressed in ppbv) of five identified sources including oil refining process (a), NG (b), combustion source (c), asphalt (d) and fuel evaporation (e), and specific compounds with high loadings in each source profile. Note that the CO in combustion source was expressed in mg m$^{-3}$.



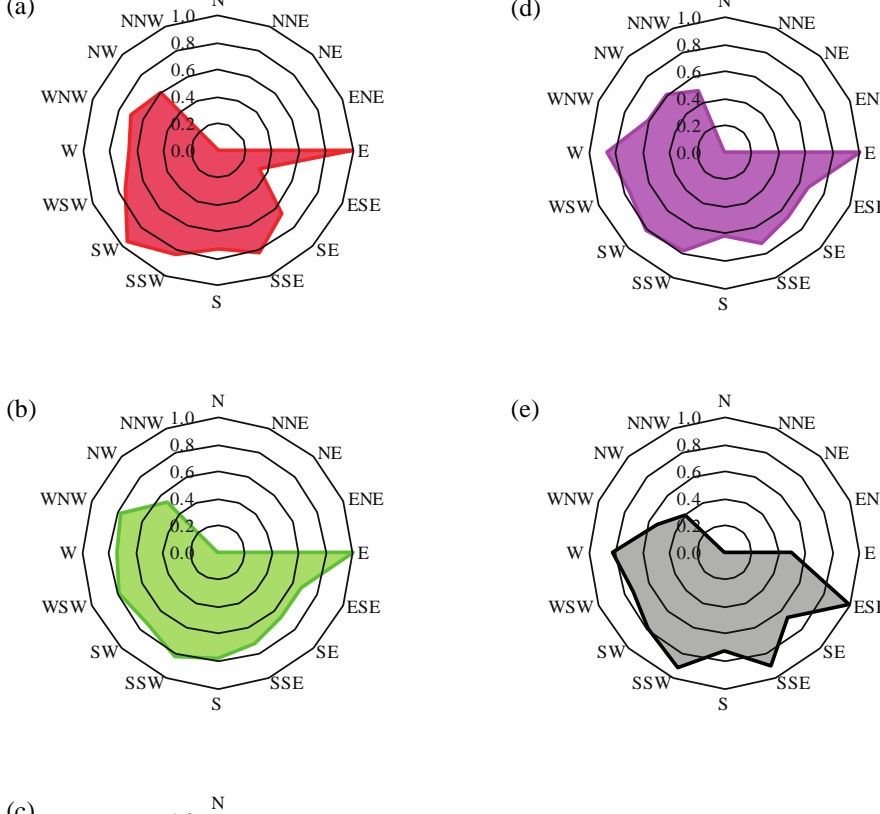

**Figure 14.** Annual conditional probability function (CPF) plots of five identified VOCs source including oil refinery (a), NG (b), combustion (c), asphalt (d), and fuel evaporation (e).







**Figure 15.** Annual weight potential source contribution function (WPSCF) maps for five identified sources derived from PMF analysis including oil refining (a), NG (b), combustion source (c), asphalt (d), and fuel evaporation (e). The black cross represents the sampling site.





**Figure 16.** Annual weight concentration weighted trajectory (WCWT) maps for five identified sources derived from PMF analysis including oil refining (a), NG (b), combustion source (c), asphalt (d), and fuel evaporation (e). The black cross represents the sampling site.



**Appendix A Detail operation of positive matrix factorization (PMF) in source apportionment of VOCs dataset.**

**A1 Data preparation**

Two files including species concentrations and uncertainty file are required to be introduced into the EPA PMF 5.0 model. The concentrations file is a matrix ($X$ matrix) of number of samples (column) plus the number of species (row), i.e., $2743 \times 20$ in this study. There are two types of uncertainty files: sample-specific and equation-based. The sample-specific uncertainty file is also a matrix with the same dimension as the concentrations matrix. The equation-based uncertainty dataset is constructed according to the method detection limit (MDL) and error fraction (%):

$$U_{ij} = \frac{5}{6} \times MDL \qquad C \leq MDL \tag{1}$$

$$U_{ij} = \sqrt{(ER \times concentration)^2 + (MDL)^2} \qquad C > MDL \tag{2}$$

Not all 56 Photochemical Assessment Monitoring Stations (PAMS) VOCs are introduced into the PMF model, there are some rules to decide which species should be included or excluded from the PMF model: (1) highly collinear species, such as propane & $n$-butane, benzene & toluene are included (Fig. A1); (2) species indicating VOC sources (i.e., acetylene is marker of combustion sources) are retained; (3) species that are highly reactive are excluded (i.e., $i$-pentene), since they are rapidly reacted away in the ambient atmosphere (Guo et al., 2011; Shao et al., 2016). Prior to the PMF model run, the retained species are firstly classified into strong, weak, and bad based on their signal-noise-ratios (S/N). Species with S/N ratios less than 0.5 are grouped into bad and into weak if S/N ratios are in the range of 0.5-1.0 (US EPA, 2014). However, S/N ratios are not useful to categorize species because all species have S/N ratios greater than 2.0 in this study. Therefore, the method limit detection, residual scale, and priority knowledge of VOCs source tracers are used. The species with percent below MDL greater than 60% are categorized as bad and excluded from the model (i.e., $trans/cis$-2-pentene, isoprene); species with percent below MDL > 50% are characterized as weak (Callén et al., 2014). Finally, nine species (ethane, propane, $n$-hexane, cyclohexane, methylcyclohexane, $n$-octane, $n$-nonane, $n$-decane, and $u$-undecane) were categorized as strong and eleven species ($i$-butane, $n$-butane, $i$-pentane, $n$-pentane, $n$-dodecane, ethylene, acetylene, benzene, toluene, $m, p$-xylene, and $o$-xylene) were characterized as weak due to its residual scale beyond 3.

**A2 The optimal number of factors.**

Choosing the optimal number of factors (P-value) in modeling is a critical question. Too many factors will result in meaningless factor profiles, while too few factors will make it difficult to segregate the mixing sources (Bressi et al., 2014). Factors ranged from 3 to 8 were tested in this study. Each model was run for 20 times with a random seed. All the Q values ($Q_{true}$, $Q_{robust}$, $Q_{except}$, and $Q_{ture}/Q_{except}$), observed verse predicted (O/P) concentrations and scaled residuals were evaluated. In theory, if the number of sources is estimated properly, the $Q_{true}$ value should be approximately to $Q_{except}$. If the number of sources is not well determined, the Q value may deviate from the theoretical value (Guo et al., 2011; Bressi et al., 2014; Baudic et al., 2016).




However, the $Q_{true}$ always deviate from the $Q_{except}$ in many cases especially in large dataset (Liu et al., 2016, 2017; Shao et al., 2016). The variation of the Q values to the number of factors is shown in Fig. A2a and the correlation coefficients between O/P values in each factor number solution is shown in Fig. A2b.

As shown in Fig. A2a, $Q_{true}/Q_{except}$ decreased substantially between 2, 3 and 4 factor solutions, indicating that a substantial
amount of the variability in the dataset was accounted for with each additional factor; for P=5, the $Q_{ture}/Q_{except}$ exhibited the minimum value; as the factor number moved from 6 to 8, the $Q_{ture}/Q_{except}$ value increased again. The Pearson correlation coefficients between observed and predicted total VOC concentrations for different factor numbers are shown in Fig. A2b, which indicating that the total VOC concentrations were well reproduced by PMF model. In addition, for the 20 individual VOCs species, the PMF mode also well reproduced the predicted concentrations, with the $R^2$ ranged from 0.42 to 0.96 (Table
A1). Therefore, we considered that the 5-factor solution was the optimum solution for this PMF analysis (Fig. A3).

### A3 Bootstrap run (BS)

After choosing the 5-factor solution, the bootstrap (BS) method was used to detect and estimate disproportionate effects of a small set of observations on the solution and also, to lesser extent, effects of rotational ambiguity. BS datasets were constructed by randomly sampling blocks of observations from the original data set (US EPA, 2014). The base run with the lowest $Q_{robust}$
was provided to map with each BS run in minimum Pearson correlation coefficient being 0.6. The number of BS was set as 100 to ensure the robustness of the statistics. In this study, the base and boot factors were matched except for two factors especially due to the factor 5 (fuel evaporation) (Table A2). The factor 1 (oil refinery) and factor 2 (NG) containing factor 5 species and factor 3 (combustion) also mapped to boot factor 5. Mapping over 80% of the factors indicated that BS uncertainties can be interpreted and the number of factors may be appropriate. Seen from the Table A2, the BS results indicated
a rotational ambiguity and F-peak should be further applied.

### A4 BS-DISP error estimation

BS-DISP estimates the errors associated with both random and rotational ambiguity. A key file containing the number of cases accepted, largest decrease in Q, number of swaps in best fit and DISP, and swaps by factor (Table A3) was used to assess the error fraction. There were 99 bootstrap cases accepted and one resample rejected proved by the largest decrease in Q being -
14. The decrease of Q was less than 1%, which indicated that the test of BS was validated and no more testing was required. It suggested that the solution was well constrained and the BS-DISP results can be reported.

Finally, the F-Peak values from -1 to 1 at 0.1 interval were used to remove the rotational ambiguity as discussed above. The F-peak bootstrap was also used to test the mapping between the base model and the F-peak runs. The results indicated that the F-peak=0.2 was the optimal solutions with all factors mapping 100% and all the species of base run within the inter quartile
range (IQR) of the BS run.





**Appendix B The calculation of PSCF and CWT**

**B1 PSCF**

The PSCF values were calculated to explore the potential geographic origins of VOC sources using the source contributions apportioned from the PMF model and the backward-trajectory. The PSCF is defined as:

$$PSCF_{ij} = \frac{m_{ij}}{n_{ij}} \tag{4}$$

where $i$ and $j$ are on behalf of the latitude and longitude, $n_{ij}$ is the total number of endpoints that fall in the $ij$-th cell, and $m_{ij}$ is defined as the number of endpoints in the same cell that exceeded the threshold criterion. The 75th percentiles of each identified source contribution (3.9 ppbv for oil refinery, 75.1 ppbv for NG, 15.3 ppbv for combustion, 1.2 ppbv for asphalt, and 33.4 ppbv for fuel evaporation) was used as the criterion value. When each grid average number of trajectory endpoint ($n_{ave}$) is three times larger than the $n_{ij}$, the uncertainty of cell is reduced by multiplying a weight function ($W_{ij}$) into the PSCF value. The weight function is expressed as:

$$WPSCF = \frac{m_{ij}}{n_{ij}} \times W(n_{ij}) \tag{5}$$

$$W(n_{ij}) = \begin{cases} 1.00, & n_{ij} > 3n_{ave} \\ 0.70, & 3n_{ave} > n_{ij} > 1.5n_{ave} \\ 0.40, & 1.5n_{ave} > n_{ij} > n_{ave} \\ 0.20, & n_{ij} > n_{ave} \end{cases} \tag{6}$$

**B2 CWT**

Since PSCF value just gives the proportion of potential sources in a grid with struggles to distinguish the pollution levels of different potential regions, a concentration-weighted-trajectory (CWT) model was employed in this study. The geographical-domain was sliced into grid cell with a resolution of 0.5 °×0.5 °. The CWT was calculated according to:

$$C_{ij} = \frac{1}{\sum_{l=1}^{M} \tau_{ijl}} \sum_{l=1}^{M} c_l \tau_{ijl} \tag{7}$$

where $C_{ij}$ represents the average weight concentrations in the grid cell (i, j), $C_l$ is the measured VOCs concentration observed on the arrival of trajectory $l$, $\tau_{ijl}$ is the number of trajectory end points in the grid cell (i, j) associated with the $C_l$ sample. The weighting function described in equation (6) was also used in the CWT analyses.



## Appendix C The detail operation of local emission and regional transport contribution

### C1 Choosing the radius to distinguish the local and regional area.

Grids within a given radius were excluded in case they could be too close to the origin to be properly distributed. Usually, the radius was set according to the area covered by first 6 h of trajectories (Bari et al., 2003; Wang et al., 2015; Wang et al., 2016). In this study, we referred to it with modification. 12 h backward trajectories were set as the radius to distinguish the local and regional area due to the following reasons:

(1) The duration of *m, p*-xylene decreasing from highest concentrations (at 02:00 LT) to lowest (14:00 LT) was 12 h (section 3.4). The atmospheric lifetime of *m, p*-xylene is about 11.8 h assuming the OH radical equals to $10^6$ rad cm$^{-3}$. The compounds with atmospheric lifetime longer than *m, p*-xylene can be transported from long distance or accumulated in local area.

(2) The endpoint of every 2h backward trajectories in the first 24 h was test to find the optimum range of "local and nearby" area (Fig. C1). As the backward time increasing from 1h to 24h, the area covered by the long air mass increased significantly. Before the first 5h, the air masses were mainly from the northwest and the east of the sampling site. From 7h to 12h, air masses from the northeast, southeast, and southwest reached the receptor site and the "shape of local area" formed. After 12h, the air mass, especial the trajectories from the west transported from long distance reached the sampling site, indicating the regional conditions.

It can be seen from the Fig. C1 that the farthest endpoint of 12h backward was about 7 degrees away from the sampling site. Therefore, the local and regional area was divided by a circle with the radius being 7 degrees (sampling site as the origin).

### C2 Raster analysis

The CWT results obtained by TrajStat software were stored in shapefile format and then introduced into Arc GIS software (10.1, Esri, US). The first step was to remove the negative value from the shapefile and then convert the shapefile into raster format. The local and regional area was extracted by a circle with radius being 7 degrees as discussed above. The inner of the circle was defined as the local while the external area was set as the regional transport and an example is shown in Fig. C2. The statistics (count, minimum, maximum, sum, mean, and standard deviation) of the extracted raster were shown in its layer properties in Arc Map. The statistics of each VOCs source in each season are summarized in Table C1. The percentage of contributions of local source and regional transport of five VOCs source in different seasons were calculated according to Eq. (6) in section 2.6.3.





**Table A1.** Pearson coefficients between the observed and predicted VOCs concentrations for 5-factor solution

| Species | Intercept | Slope | SE[a] | $R^2$ |
|---|---|---|---|---|
| ethane | 1.09 | 0.90 | 7.69 | 0.96 |
| propane | 0.25 | 0.98 | 5.96 | 0.94 |
| *i*-butane | 0.42 | 0.86 | 4.27 | 0.88 |
| *n*-butane | 0.76 | 0.82 | 6.06 | 0.84 |
| *i*-pentane | 2.50 | 0.71 | 7.57 | 0.50 |
| *n*-pentane | 0.69 | 0.75 | 4.20 | 0.77 |
| *n*-hexane | 0.24 | 0.84 | 1.05 | 0.91 |
| cyclohexane | 0.11 | 0.82 | 0.37 | 0.92 |
| methylcyclohexane | 0.14 | 0.81 | 0.43 | 0.93 |
| *n*-octane | 0.06 | 0.84 | 0.21 | 0.93 |
| *n*-nonane | 0.04 | 0.71 | 0.10 | 0.85 |
| *n*-decane | 0.05 | 0.68 | 0.08 | 0.79 |
| *n*-undecane | 0.04 | 0.70 | 0.09 | 0.71 |
| *n*-dodecane | 0.17 | 0.27 | 0.29 | 0.49 |
| ethylene | 0.25 | 0.50 | 0.75 | 0.53 |
| acetylene | 0.57 | 0.15 | 0.93 | 0.42 |
| benzene | 0.36 | 0.33 | 0.56 | 0.42 |
| toluene | 0.32 | 0.24 | 0.48 | 0.37 |
| *m, p*-xylene | 0.14 | 0.48 | 0.32 | 0.57 |
| *o*-xylene | 0.05 | 0.36 | 0.07 | 0.55 |

[a] SE=standard error





**Table A2.** Mapping of bootstrap factors to base factors

|               | Factor 1 | Factor 2 | Factor 3 | Factor 4 | Factor 5 | Unmapped |
|---------------|----------|----------|----------|----------|----------|----------|
| Boot Factor 1 | 100      | 0        | 0        | 0        | 0        | 0        |
| Boot Factor 2 | 0        | 100      | 0        | 0        | 0        | 0        |
| Boot Factor 3 | 2        | 3        | 95       | 0        | 0        | 0        |
| Boot Factor 4 | 0        | 0        | 0        | 100      | 0        | 0        |
| Boot Factor 5 | 14       | 20       | 0        | 0        | 66       | 0        |



**Table A3.** BS-DISP Diagnostics.

| | | | | | |
|---|---|---|---|---|---|
| # of Cases Accepted: | 99 | | | | |
| % of Cases Accepted: | 99% | | | | |
| Largest Decrease in Q: | -14.10 | | | | |
| %dQ: | -0.014 | | | | |
| # of Decreases in Q: | 1 | | | | |
| # of Swaps in Best Fit: | 0 | | | | |
| # of Swaps in DISP: | 0 | | | | |
| Swaps by Factor: | 0 | 0 | 0 | 0 | 0 |




**Table C1.** The statistics of each VOCs source in each season obtained by raster analysis

|  |  | Oil refinery | | NG | | Combustion | | Asphalt | | Fuel evaporation | |
| --- | --- | --- | --- | --- | --- | --- | --- | --- | --- | --- | --- |
|  |  | Local | Regional | Local | Region | Local | Region | Local | Region | Local | Region |
| Autumn | Minimum | 0.016 | 0.00055 | 0.083 | 0.005 | 0.021 | 0.019 | 0.022 | 0.0032 | 0.012 | 0.0027 |
|  | Mean | 2.68 | 0.48 | 41.15 | 4.63 | 7.6 | 1.16 | 0.98 | 0.17 | 17.43 | 2.9 |
|  | Sum | 816 | 449 | 12758 | 4347 | 2356 | 1098 | 305 | 160 | 5315 | 2468 |
|  | Number | 304 | 928 | 310 | 939 | 310 | 943 | 310 | 944 | 305 | 851 |
| Winter | Minimum | 0.017 | 0.0007 | 0.62 | 0.089 | 0.092 | 0.092 | 0.0052 | 0.00043 | 0.13 | 0.0025 |
|  | Mean | 1.41 | 0.32 | 36.1 | 7.82 | 12.34 | 2.93 | 0.35 | 0.076 | 18.51 | 4.21 |
|  | Sum | 492 | 319 | 12707 | 8411 | 4333 | 3147 | 121 | 79 | 6406 | 4421 |
|  | Number | 348 | 1010 | 352 | 1076 | 351 | 1073 | 345 | 1036 | 346 | 1049 |
| Spring | Minimum | 0.0027 | 0.000094 | 0.11 | 0.064 | 0.48 | 0.033 | 0.0066 | 0.0041 | 0.002 | 0.0015 |
|  | Mean | 1.05 | 0.19 | 16.71 | 2.86 | 2.6 | 0.5 | 0.49 | 0.093 | 7.66 | 1.91 |
|  | Sum | 360 | 180 | 6068 | 3175 | 945 | 553 | 177 | 103 | 2764 | 1840 |
|  | Number | 343 | 954 | 363 | 1112 | 363 | 1111 | 363 | 1109 | 361 | 964 |
| Summer | Minimum | 0.00044 | 0.0011 | 0.036 | 0.021 | 0.01 | 0.0021 | 0.0082 | 0.00022 | 0.0097 | 0.00028 |
|  | Mean | 1.92 | 0.29 | 19.57 | 2.6 | 2.63 | 0.4 | 0.51 | 0.1 | 9.98 | 1.87 |
|  | Sum | 556 | 242 | 5714 | 2250 | 776 | 343 | 149 | 89 | 2923 | 1513 |
|  | Number | 289 | 845 | 292 | 865 | 295 | 863 | 293 | 876 | 293 | 808 |
| Annual | Minimum | 0.0036 | 0.000094 | 0.0356 | 0.011 | 0.0645 | 0.0019 | 0.0023 | 0.00043 | 0.0036 | 0.0015 |
|  | Mean | 2.16 | 0.41 | 35.26 | 6.52 | 8.08 | 1.87 | 0.73 | 0.16 | 16.44 | 3.8 |
|  | Sum | 953 | 821 | 15973 | 13338 | 3653 | 3805 | 328 | 330 | 7399 | 7225 |
|  | Number | 442 | 1959 | 453 | 2047 | 452 | 2035 | 450 | 2014 | 450 | 1899 |





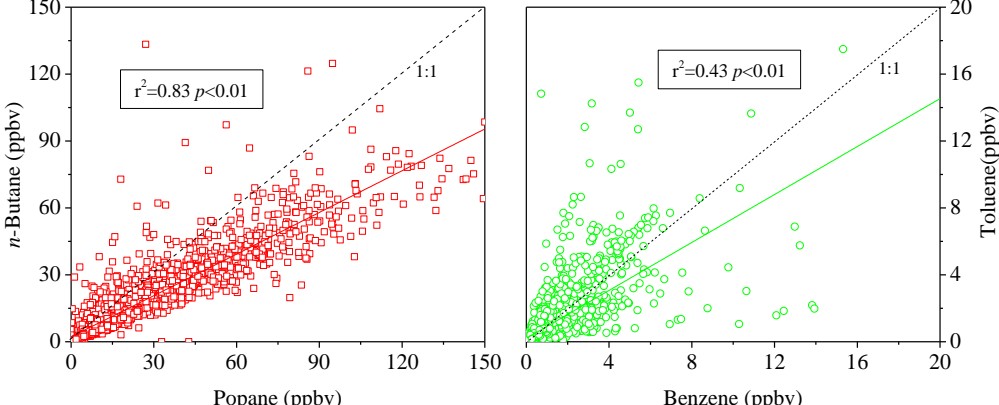

**Figure A1.** Samples of highly collinear species.





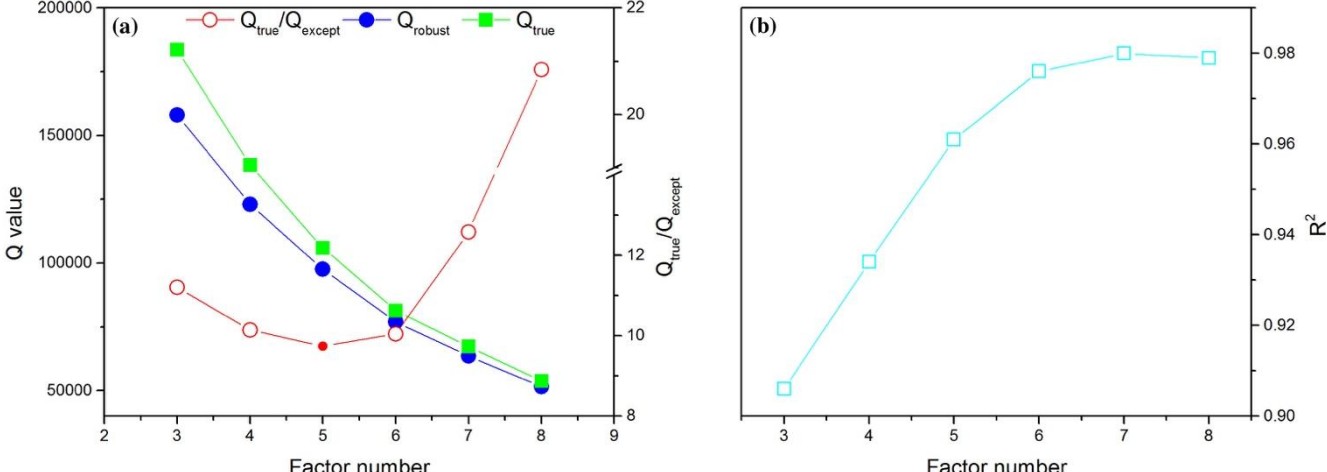

**Figure A2.** $Q_{true}/Q_{except}$, $Q_{robust}$, $Q_{true}$ plotted against the number of factors used in the positive matrix factorization (PMF) solution.





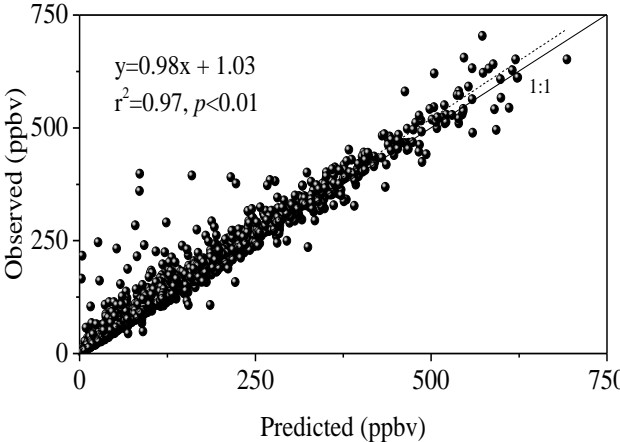

**Figure A3.** Scatter plots between the total predicted and observed VOC concentrations based on the 5-factor PMF solution.



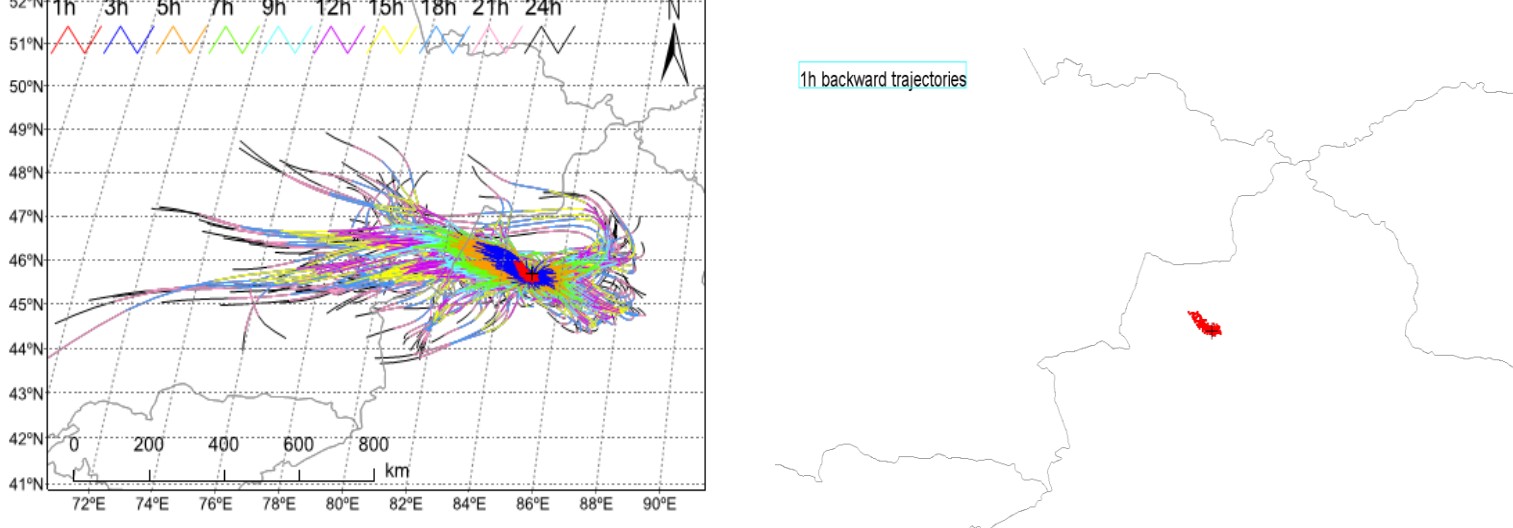

**Figure C1.** The region covered by first 24h backward trajectories. Left: the domain covered by the trajectories. Right: the dynamic variation of backward trajectory from 1h to 24h. (Double click to view the GIF)



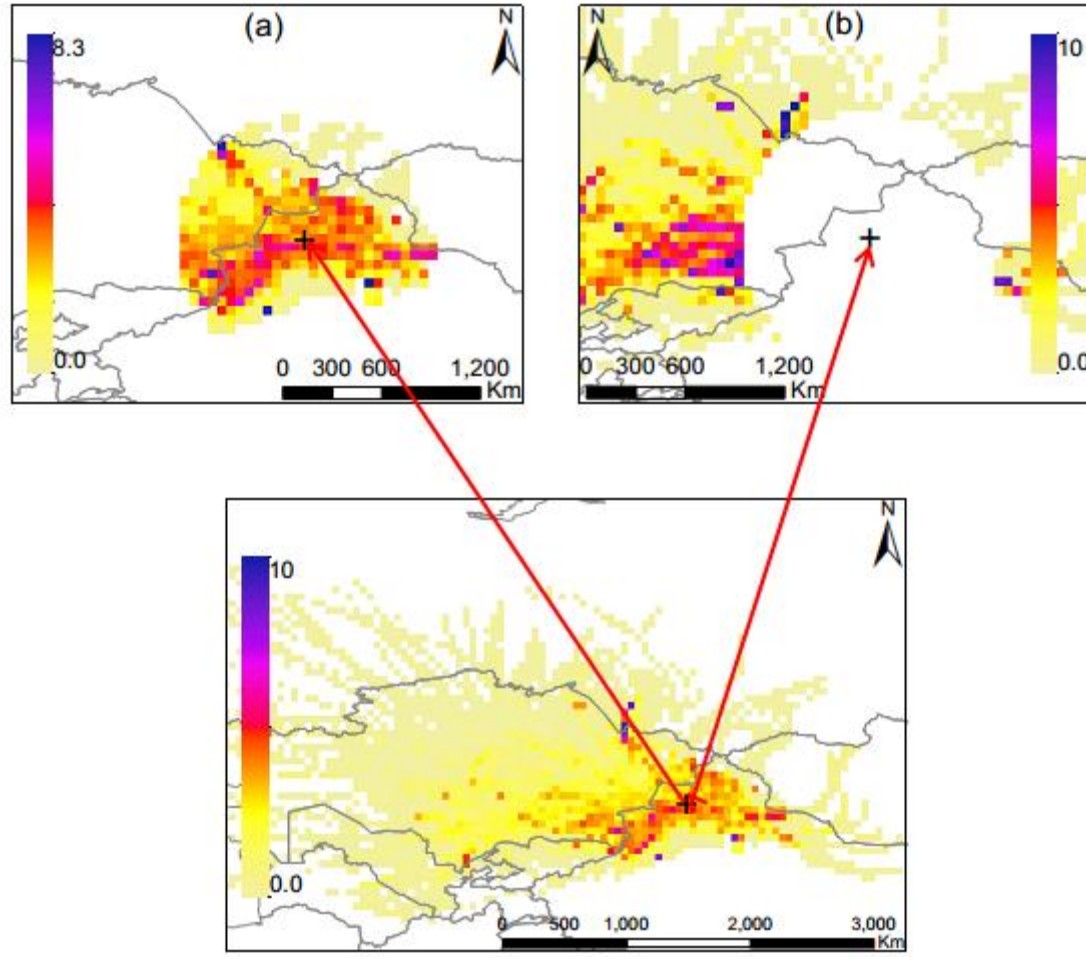

**Figure C2.** Example of (a) local area and (b) domain of regional transport range extracted by a circle with the radius being seven degrees. The black cross represents the sampling point and set as the original point.