# Peer review of "One year monitoring of volatile organic compounds (VOCs) from an oil and gas station in northwest China"

_Atmospheric Chemistry and Physics, 2017_

## Referee Comment (RC1) · Anonymous Referee #2 · 30 Nov 2017

This paper reported one year VOC measurements in high frequency at oil-gas station area in China. High frequency VOC data enable to show diurnal variations of VOCs in the sampling site and strong influence of meteorological condition for VOC concentrations was elucidated. Using PMF analysis, VOC emission was successfully categorized into five sources. Further, CPF analysis gave more information about the direction of the VOC emission sources. Also backward trajectory based analysis (PSCF, CWT) were applied.

[ General comments ] Similar VOCs measurement and analysis were already demonstrated in urban area, but the detailed VOC observation at oil-gas station area will be

important in view point of source area information. Because strong VOC emission sources are close to the measurement site, the analyzed results would be simpler and seems to be reasonable. But I am not sure about the validity of PSCF and CWT analysis in this measurements. When there is a strong source very close to the measurement site (like 2 km from Oil refinery and 6km from Oil-gas field in Fig 1c), the backward trajectories passed the near emission source will be counted as high concentration over its whole trajectory rout. Therefore, the raster analysis (distinguish local and regional area) would be reasonable trial. (But I am afraid that the influence of very close emission sources can not be excluded, because that 12h radius will be still long and concentration of each grid was estimated from CWT.)

[ Detailed comments ] Fig1 : What is 2 type green area in Fig 1(a)? What is blue area in Fig1(b) ? Exact location of the sampling site is difficult to see in Fig1 (b) (because of blue color). Is the sampling site located in Region 1? What is the rose figure in Fig1c ? Wind speed or VOC concentration from each direction? If this rose figure shows wind direction, figure caption will be incorrect ("Northwesterly and northeasterly wind prevailed").

Page4 L3 : PLOT column will be used for C2-C4 separation in GC. For C2-C4 trap, some absorbent (tenax etc) would be used. Please check the explanation about VOC measurement system.

Table 1 : Please explain "MDL" (method detection limit). "KOH" will be better to write in small characters.

Fig3 : Is the colors in Fig3(e) same as Fig3(a)-(d)? (In Fif3(e), blue (acetylene?) seems larger contribution. )

Fig6 and text: Are O3, BLH etc 24hour average or only daytime average? There are strong diurnal variations in BLH, O3 and VOCs, so these plot would be different when you use whole day or only daytime data.

Fig12 (d) : Is it correct "F1-F4" ? (Is it "CO-F4" ?)

Fig13 : Please explain MCH and MCP. In the right axis of (a),(b),(e), "/" will be better to show as ",". They are not ratio, but just concentration. "Cyclohexane" etc would be better to show as dot and line.

P12 L24-2: Diurnal pattern of acetylene seems to be different. (Acetylene did not decrease during daytime.)

P15 L20-22 : "Northeast to Southwest" Is this "Southeast to Southwest"?

P15 L23-24 : Fig.14(b), highest peak is in east direction.

---

## Referee Comment (RC2) · Anonymous Referee #3 · 6 Dec 2017

This paper describes one year continuous monitoring of VOCs around an oil-gas region in northwest China in order to clarify atmospheric behavior of VOCs in such region. The authors revealed temporal variations such as seasonal and diurnal variations of VOCs around the oil-gas region and analyzed factors of such variations. In addition, they performed source analyses of VOCs and discussed source of VOCs in this region quantitatively.

General comments:

As the authors mentioned, VOCs are main precursors of tropospheric ozone and it is important to clarify atmospheric behavior of VOCs. Examples of VOC observations in

oil-gas regions are low, especially; there are few continuous observations of VOCs with high time resolution. The authors supply valuable data and information. In addition, the authors conducted quantitative source analyses of VOCs. I recommend this paper to be published in Atmospheric Chemistry and Physics. However, I found several dubious points in this paper. The authors should revise appropriately.

Specific comments:

The authors performed several discussions using $NO_2$. Why do the authors use $NO_2$ instead of $NO_x$? I think it is preferable to use $NO_x$ instead of $NO_2$ (or both $NO_2$ and $NO_x$) for many of such discussions. The authors would observe NO and $NO_2$ because they used a TEI $NO_x$ analyzer based on a chemiluminescence method.

$NO_2$ and $NO_x$ concentrations measured by a TEI $NO_x$ analyzer are not accurate because of interferences of descendant spices of $NO_x$ such as $HNO_3$ and PANs. The authors should evaluate such interferences. Especially, organic nitrates could interfere the values of $NO_2$ concentrations obtained by a TEI $NO_x$ analyzer under high concentrations of large hydrocarbons.

On page 9, lines 6-7, "It should be noted that VOCs... as well as BLH.": I think $NO_2$ concentrations are controlled solar UV and concentrations of NO and $O_3$ as well as BLH, but are VOCs controlled concentrations of NO and $O_3$? (I don't think so.) The authors should discuss this matter separating VOCs and $NO_2$.

Table 1: The authors should explain $r^2$.

[Figure]

---

## Referee Comment (RC3) · Anonymous Referee #4 · 8 Dec 2017

This paper entitled "One year monitoring of volatile organic compounds (VOCs) from an oil-gas station in northwest China" utilized a unique dataset to analyze the differences between the VOC concentrations, compositions, source contributions in an oilgas station and other urban areas and industrials. The results seem to be interesting with unique characteristics of VOC compositions and sources in this kind of areas. Based on one-year online monitoring of VOC concentrations, the PMF model was successfully employed to source apportionment and the different timescale variations of different source contributions were discussed. The PSCF and CWT method were also employed to investigate the potential geographic origins of VOCs. A new method based on CWT was proposed to attempt to distinguish the local and regional contributions. I

suggest this paper can be accepted after minor revision and addressing my questions. The specific comments are listed as follows:

1. P1 Line 15 the sentence "the ambient VOCs from fifty-six Photochemical Assessment Monitoring Stations (PAMS) VOCs were continuously measured for an entire year (September 2014-August 2015) by a set of on-line monitor system from an oil-gas station in northwest China." confused me. Pls make it clear. 2. P1 Line 31: How about replace the keywords "source region and local-regional contribution" to local-regional contribution? 3. P2 Line 6: Insert references after "air quality". 4. P2 Line 25 Insert references or link. 5. P4 Line  $2 \sim 5$ : Please check and make sure the analysis method is correct. 6. P4 Line 8. Technic errors. The PAMS standard gases contain 57 VOC species, including alkane (30), alkene (9), alkene (acetylene), and aromatic (17). 7. P6 Line 7: The author mentioned that the trajectories were mainly originated from the northwest during the whole sampling period. However, the wind rose (Fig. 1c) indicated the northeasterly winds prevailed in P5 Line 2. How to explain the difference? 8. P7 Line 15: You mean  $33 \pm 33$  ppby? 9. P9 Line 12: The concentrations of O3 precursors decreased and O3 increased? 10. P9 Line 16~23: When discuss the effects of BLH and photochemical reactions on VOC concentrations in summer and winter. I suggest more statistical method such as ANOVA analysis can be used to test the differences were significant or not. 11. P11 Line 18-19 & P12 Line 8~10: The author compared the source contributions in different seasons using the percentage contributions (%) and volume contributions (ppbv) and it's paradoxical using both methods. For instance, in P12 Line  $8 \sim 10$ , the percentage contribution in spring was the highest, however, the volume contribution was the lowest among the four seasons. How to explain or avoid? 12. P15 Line 18: Highest CPF values of oil refinery was found in the east direction (Fig. 14a, 14b, and 14d). This sentence confused me. 13. P25 Table 1: There are some mistakes such as an extra line under n-decane. 14. P28 Table 3: I am wondering why the average value of four different seasons does not equal to annual value? 15. P35 Figure 7 and P41 Figure 13: Due to the time resolution of meteorological parameters, BLH are three hours, while the time resolution of trace gases is one hour as the author

ACPD
mentioned. I suggest that the Pearson correlation can be conducted to give a more statistical reliable relation between VOC concentrations, different source contributions and trace gases.

**ACPD**

---

## Referee Comment (RC4) · Anonymous Referee #1 · 13 Dec 2017

**Review of "One year monitoring of volatile organic compounds (VOCs) from and oil-gas station in northwest China.**

The authors of this paper have collected real-time high-resolution VOC concentration data for 56 VOCs in a location in China that is near high density oil and gas and petrochemical activity. VOC concentration data from oil and gas rich regions have been collected and published before, but the data are novel in that they were collected at a region where no such studies that targeted oil and gas emissions were conducted. The data and some of the discussion/comparison presented in the paper are valuable, however, there are a lot of redundant information in the paper and some parts that need additional clarification. There is also a lot of grammatical errors in the paper that need to be addressed before the paper is ready for publication. I recommend publication of the paper after major revisions.

**General comments:**

**Title:**

Throughout the paper, the term oil-gas or oil and natural gas have been used. Unless the authors are specifically directing attention to wet gas (which is a combination of oil and gas from the same well), they should use the term oil and gas.

The abstract needs major revisions (see Specific comments).

Section 3.3: The authors do a lot of correlation comparisons in the paper. This is noted in the abstract too. A correlation (r=0.19) is not a good correlation. Furthermore, the authors do not discuss what this actually would mean in the physical sense. This is noted in the abstract (for correlation with pressure) and page 8 and 9 (Section 2.2). What exactly does a negative correlation of r=-0.29 between VOCs and temperature? Which VOCs were correlated? Were some better or worse? Are we looking at total VOCs? If yes, this does not make sense as some VOCs are broken down by UV or other atmospheric constituents in place because of sunlight. This section needs a lot of revising and some deletion of information that is not relevant or useful.

The authors have collected data in China and the value of the data is that such data have not been collected in this area. The way they introduce this concept in the introduction is very-very confusing. I suggest the authors discuss the studies done in China in more detail (when justifying this research) and then when comparing their results with other oil and gas basins, discuss location other than China where there have been a multitude of such studies and data presented.

Was there a reason the authors did not use HYSPLIT for the back-trajectory analysis?

Throughout the paper authors should clarify their comments about other studies and data availability to show when they are talking about China vs other regions and locations.

Sections 2.3-2.5 should be re-written to be more concise and then combined. For example, PMF is a model widely used, so the amount of information given on the model is not useful in this paper.

In the photochemical and diurnal pattern discussion, the authors note that 8-14 is the time when the ratios are reduced because of photochemical removal due to enhancement of ambient temp. Ambient temp is not responsible for photochemical reaction. The increased in reactive radicals such as OH is. This should be corrected throughout this discussion.

What are the uncertainties associated with authors calculations and discussion in section 3.5? This is important as the authors want to show such specific contribution sources in a region where the sources are so close to each other. Also, acetylene is a strong marker for combustion, but it has known to be released from some industrial and oil and gas sources too. It is OK to use is as such, but this point should be addressed in the paper.

The conclusion section should be changed to summary as it is just summarizing the results not adding any conclusions. The conclusions of the paper should relate only to this area. Also, when discussing all VOCs, are the benzene concentrations in this study higher than those in the middle of a high traffic urban area? Using the term VOC to encompass everything discussed here is not a correct comparison. Isn't BLH a meteorological condition? So, what is the difference between it and the met conditions affecting monthly and daily VOC variations? #2 in conclusions does not add anything and should be removed.

The figures in this paper are not well prepared. There is too much going on in each figure (sometime irrelevant information). For example, for diurnal pattern figures the addition of day and night color only add information that are not useful.

**Specific comments:**

Page 1, Line 15: The authors do not discuss the atmospheric behaviors of the VOCs in the paper. The closest they come to this is when they discuss the ozone formation potential of the VOCs. Most of the paper is on the VOC concentrations and correlations. The term behaviors should be deleted.

Page 1, Line 17-18: The authors note that the concentrations of VOCs in this study were 1-50 times higher than those measured in many other urban and industrial regions. This statement is vague. It should either be more specifically addressed (what VOCs, which other studies) or taken out.

Page 1, Line 21-22: After Section 3.3 is revised this sentence should be deleted. First, because the correlations were not very strong; second, because correlation (even if present) does not imply causality or influence.

Page 1, Line 24: What do the authors mean by asphalt? Is this an asphalt making factory or just asphalt on the roads?

Page 1, Line 25-26: What do the authors mean by "Clear temporal variations differed from one source to another was observed…" temporal variations of what?

Page 1, Line 28: define CWT

Page 1, Line 28: What do the authors mean by "local emissions"? Are these the emissions from the oil and gas field? From the urban location?

Page 1, Line 29-30: Even though the data presented here is a good first step to look at the oil and gas emissions in this region, it is far from filling the gaps. Please revise this sentence to reflect what the paper actually does.

Page 2, Line 5-6: The authors are implying that the reason for the detrimental effect of VOCs to human health is their activity as ozone precursors. This is not correct. It is a part of the picture. VOCs such as benzene are carcinogens and bad for human health, independent of their activity as ozone precursors.

Page 2, Line 7: Indicate the region of discussion because the previous studies in other regions have certainly focused on VOCs from oil and gas.

Page 2, Line 15-16: This sentence is not clear and needs revision. What do the authors mean by "depended on processing stages"? Are the authors discussing the processing stages (drilling, fracking, production) or something else?

Page 2, Line 17-18: Gilman et al found that oil and gas emissions contribute strongly to some VOCs not all VOCs. Please correct this sentence to reflect that.

Page 3, Line 9-10: How are the authors claiming that this specific study "contributes to establish the control measures of VOCs at this type of region"? What do they mean by this type of regions?

Page 3, Line 15-16: Where did the authors get the production and deposit size data? Please cite source

Page 4, Line 26: The authors note that 3-hour met data have been collected but then in Fig 2 they show hourly data? How was this done?

Page 4, Line 29: This should be in results not in methods.

Page 10, Line 1-5: The difference is not only due to fuel evaporation. Based on the information presented here there is an urban area nearby and may contribute to the i/n pentane ratios. If this is not the case, the authors should add clarification.

Page 10, Line 19-35: This section should be re-written for clarity.

Page 11, Line 10: What is asphaltic?

**Examples of grammatical errors (these are not all the errors in the paper. The paper should go through a rigorous edit for grammar before publication).**

Page 1, Line 13: add for between important and energy

Page 1, Line 14: replace abundant with a grammatically correct adjective

Page 1, Line 17: revise "set of monitor system"

Page 1, Line 25-26: revise for grammar

Page 2, Line 2: revise "volcanos eruption"

Page 2, Line 13: replace intensive with a grammatically correct adjective

Page 3, Line 2: revise "few have concerned the local and regional source…"

Page 3, Line 3: replace long time with long-term

Page 3, Line 5: Replace researches with grammatically correct term

Page 3, Line 16: Please correct sentence so that "It" does not refer to 126 petrochemical plants but the area.

Page 3, Line 23: revise deeply

Page 3, Line 28: The authors make it sound like there were 56 monitoring sites.  I assume this refers to the number of VOCs?  Please revise sentence to reflect that.

Page 7, Line 13-17: Revise sentences

There are much more, which made reading the manuscript very difficult.  The authors should have used a technical writer.

---

## Author Comment (AC1) · 25 Jan 2018

**Response to Referee #1**

Thank you very much for your thoughtful and constructive comments on our manuscript. We have revised the manuscript accordingly. The detailed responses are given below point by point (in blue), and the revised manuscript is shown in red.

Review of "One year monitoring of volatile organic compounds (VOCs) from and oil-gas station in northwest China. The authors of this paper have collected real-time high-resolution VOC concentration data for 56 VOCs in a location in China that is near high density oil and gas and petrochemical activity. VOC concentration data from oil and gas rich regions have been collected and published before, but the data are novel in that they were collected at a region where no such studies that targeted oil and gas emissions were conducted. The data and some of the discussion/comparison presented in the paper are valuable, however, there are a lot of redundant information in the paper and some parts that need additional clarification. There is also a lot of grammatical errors in the paper that need to be addressed before the paper is ready for publication. I recommend publication of the paper after major revisions.

We thank the referee for the constructive evaluation and improving suggestions which were considered in revising the manuscript.

General comments:

Throughout the paper, the term oil-gas or oil and natural gas have been used. Unless the authors are specifically directing attention to wet gas (which is a combination of oil and gas from the same well), they should use the term oil and gas.

Thanks for the referee's rigorous suggestion. We confirm that the oil and gas in the study area is not from the same well. Therefore, the term oil and gas instead of oil-gas is used in the revised manuscript.

The abstract needs major revisions (see Specific comments).

We have revised the abstract.

Oil and natural gas are important for energy supply around the world. The exploring, drilling, transportation and processing in oil and gas regions can release a lot of volatile organic compounds (VOCs). To understand the VOC levels, compositions and sources in such region, an oil and gas station in northwest China was chosen as the research site and fifty-seven VOCs designed as the photochemical precursors were continuously measured for an entire year (September 2014–August 2015) using an on-line monitoring system. The average concentrations of total VOCs were $297 \pm 372$ ppbv and the main contributor was alkanes, accounting for 87.5% of the total VOCs. According to the propylene-equivalent concentration and maximum incremental reactivity methods, alkanes were identified as the most important VOC groups for the ozone formation potential. Positive matrix factorization (PMF) analysis showed that the annual average contributions from natural gas, fuel evaporation, combustion sources, oil refining process and asphalt (anthropogenic and natural sources) to the total VOCs were $62.6 \pm 3.04\%$, $21.5 \pm .99\%$, $10.9 \pm 1.57\%$, $3.8 \pm 0.50\%$ and $1.3 \pm 0.69\%$, respectively. The five identified VOC sources exhibited various diurnal variation patterns due to their different emission patterns and the impact of meteorological parameters. Potential source contribution function (PSCF) and contribution weighted trajectory (CWT) models based on backward trajectories indicated that the five identified sources had similar geographic origins. Raster analysis based on CWT analysis indicated that the local emissions contributed 48.4–74.6% to the total VOCs. Based on the high-resolution observation data, this study clearly described and analyzed the temporal variation of VOC emission characteristics at a typical oil and gas filed, which exhibited different atmospheric behaviors compared with that in urban and industrial areas.

Section 3.3: The authors do a lot of correlation comparisons in the paper. This is noted in the abstract too. A correlation (r=0.19) is not a good correlation. Furthermore, the authors do not discuss what this actually would mean in the physical sense. This is noted in the abstract (for correlation with pressure) and page 8 and 9 (Section 2.2). What exactly does a negative correlation of r=-0.29 between VOCs and temperature? Which VOCs were correlated? Were some better or worse? Are we looking at total VOCs? If yes, this does not make sense as some VOCs are broken

down by UV or other atmospheric constituents in place because of sunlight. This section needs a lot of revising and some deletion of information that is not relevant or useful.

Thanks for your constructive comments. The physical meaning of the relationship between total VOCs and meteorological data are discussed in the manuscript. In addition, the relations between some VOC species (the representative species of VOCs groups, i.e., ethane, ethylene, acetylene and benzene) were explored by Pearson correlation analysis (Table S2) and we find that meteorological condition have different impacts on different VOCs species. Some irrelevant information is also deleted. We revised this section as shown below:

Fig. 6 shows the temporal variation of ethane, ethylene, acetylene and benzene in different timescales. Though differences exist, the selected compounds broadly represent the respective alkanes, alkenes, alkynes and aromatics (Lyu et al., 2016). Significant differences were found between the meteorological parameters in different seasons ($p < 0.01$) and the highest concentrations of these species in winter were observed. The seasonal variation of VOCs is controlled by meteorological conditions, photochemical activities and source emissions. The highest values in winter was due to inhibited photochemical activities under suppressed dispersion conditions and low temperature. For instance, all these species were negatively correlated to BLH with higher VOCs levels under lower BLH (Fig. 6). The wind speed and temperature were also found negatively correlated to VOCs concentrations and ethylene showed the highest negative correlation to these parameters (Table S2). The less photochemical reactions also resulted in high concentrations in winter, which was proved by negative correlation between VOCs and $O_3$ (Table S2). Additional sources (i.e., combustion) may be also present, in view of the obvious increase of acetylene (Fig. 6c) from summer (0.87 ± 1.00 ppbv) to winter (10.5 ± 8.51ppbv). Conversely, the high temperature, wind speed and BLH favor the dilution and dispersion of ambient VOCs and the photochemical depletion in summer. Therefore, lowest VOCs concentrations were observed in summer.

The diurnal variations of VOCs and trace gases ($NO_2$ and $O_3$) related to photochemical reaction are shown in Fig. 7. The VOCs had a reverse trend with $O_3$ ($r = -0.82$, $p < 0.01$). The lower BLH and less photochemical activities resulted in peak values for VOCs and low $O_3$ concentrations before sunrise (6:00 local time). After sunrise, with the initiation of photochemical oxidation and the increasing of BLH, the concentrations of VOCs decreased while the $O_3$ increased rapidly. The minimum of VOCs and occurred at about 12:00–14:00 LT was resulted from both dispersion or dilution conditions and photochemical reactions (with highest $O_3$ concentrations at l4:00 LT) in the afternoon. The diurnal variation of $NO_2$ was controlled by BLH, $O_3$ and photochemical reactions (i.e., OH radical) and showed a double peak. The similar diurnal patterns of different atmospheric lifetime compounds including ethane, ethylene, acetylene and benzene (the most abundant contributors to its categories) were also found (Fig. S4). To better understand the effects of BLH and photochemical reactions on VOCs, the diurnal variations of VOCs, BLH and $O_3$ in winter and summer were analyzed (Fig. 7b, c). VOC concentrations in winter ($213 \pm 97.7$ ppbv) were significantly higher than those in summer ($130 \pm 100$ ppbv). However, the VOCs in summer and winter decreased by 8.3 times and 2.3 times, respectively, from maximum to minimum. This was due to the BLH increased by 8.2 times in summer while the BLH in winter only increased by 2.3 times. The effects of photochemical reactions on VOCs in two seasons were comparable, which was explained by similar $O_3$ increment in winter (0.78 times up) and summer (0.71 times up). Therefore, we can conclude that the role of BLH variation was more important than the photochemical reaction for the diurnal variation of VOCs.

Table S2 Pearson correlation between VOCs, meteorological data and trace gases (O3 and NO2)

| Species | Temperature | Wind speed | BLH | O3 | NO2 |
|---|---|---|---|---|---|
| Ethane | −0.393** | −0.436** | −0.524** | −0.414** | 0.633** |
| Ethylene | −0.590** | −0.530** | −0.653** | −0.500** | 0.642** |
| Acetylene | −0.541** | −0.370** | −0.481** | −0.197** | 0.321** |
| Benzene | −0.440** | −0.392** | −0.473** | −0.377** | 0.418** |
| VOCs | −0.286** | −0.391** | −0.444** | −0.361** | 0.603** |

**. Correlation is significant at the 0.01 level (2-tailed).

*. Correlation is significant at the 0.05 level (2-tailed).

The authors have collected data in China and the value of the data is that such data have not been collected in this area. The way they introduce this concept in the introduction is very-very confusing. I suggest the authors discuss the studies done in China in more detail (when justifying this research) and then when comparing their results with other oil and gas basins, discuss location other than China where there have been a multitude of such studies and data presented.

Thanks for this comment, we have revised the expression accordingly.

Was there a reason the authors did not use HYSPLIT for the back-trajectory analysis?

Due to our vague description, which made the readers confusing the method for back-trajectory analysis. In fact, the back-trajectory analysis was conducted using HYSPLIT, which was run by TrajSta-Meteoinfo plugin (available at http://www.meteothinker.com). Therefore, we revise our description in the manuscript.

Throughout the paper authors should clarify their comments about other studies and data availability to show when they are talking about China vs other regions and locations.

Thanks for your suggestion, comments and data availabilities about other studies have been added in the revised manuscript.

Previous studies in China mainly focused on the measurements of VOCs in urban agglomerations such as the Pearl River Delta (PRD) region (Tang et al., 2007; Liu et al., 2008; Cheng et al., 2010; Ling et al., 2011), Yangtze River Delta (YRD) region (An et al., 2014; Li et al., 2016; Shao et al., 2016) and Beijing-Tianjin-Hebei (BTH) region (Li et al., 2015) and key megacities including Beijing (Song et al., 2007; Wang et al., 2010; Yuan et al., 2010), Shanghai (Cai et al., 2010; Wang, 2014), Guangzhou (Zou et al., 2015) and Wuhan (Lyu et al., 2016). These studies found that vehicle emission and solvent usage contributed most to the ambient VOCs in urban areas. A few studies were also

conducted in industrial areas (An et al., 2014; Wei et al., 2015; Shao et al., 2016) and petrochemical industrial regions with a lot of VOC emissions (Lin et al., 2004; Wei et al., 2015; Jia et al., 2016; Mo et al., 2017). These studies found that the VOC sources and compositions are complex due to the different emission and atmospheric processes (Warneke et al., 2014). However, the research conducted in oil and gas area in China is still limited while the VOCs emission characteristics in this type of regions are common around the world (Buzcu-Guven and Fraser, 2008; Simpson et al., 2010; Rutter et al., 2015; Bari et al., 2016). For instance, Leuchner and Rappenglück. (2010) found that natural gas/crude oil sources contributed most to the VOC emissions in Houston. Gilman et al. (2013) found that oil and gas emissions strongly contribute to the propane and butanes in northeast Colorado. Therefore, study concerning VOC emission characteristics studies in oil and gas area in China is very important.

Sections 2.3-2.5 should be re-written to be more concise and then combined. For example, PMF is a model widely used, so the amount of information given on the model is not useful in this paper.

Thanks for your constructive suggestion. Section 2.3-2.5 has been combined to be more concise.

**2.3 Data sources and analysis**

**2.3.1 Meteorological parameters and air pollutants**

Other dataset such as the three-hour resolution meteorological parameters (atmospheric pressure (P), temperature (T), relative humidity (RH), wind speed (WS) and direction (WD)) were collected from the Meteomanz (www.meteomanz.com) and are shown in Fig. 2. The boundary layer height (BLH) was computed every three hours each day through the NOAA's READY Archived Meteorological website (http://www.ready.noaa.gov/READYamet.php).

The hourly CO, $NO_2$, $O_3$, $SO_2$, ambient inhalable particles ($PM_{10}$) and fine particles ($PM_{2.5}$) were measured using an ambient air quality continuous automated monitors (TH-2000 series, Wuhan-Tianhong Instrument Co., Ltd, China) and the data were acquired from the Qingyue Open Environmental Data Center (https://data.epmap.org). It should

be noted that the $NO_2$ ($NO_2 = NO_x$–NO) concentrations were in fact overestimated. This is because some oxidized reactive nitrogen that is converted by the molybdenum during the $NO_x$ measurement while the NO measurement is accurate using the chemiluminescence technic. Therefore, the $NO_2$ concentrations discussed below are considered greater than the actual values (Dunlea et al., 2007; Zou et al., 2015). According to the ambient air quality standards-Ⅱ (GB/3095-2012), the main air pollutants were $PM_{10}$ and $PM_{2.5}$ in winter and $NO_2$ in autumn (Fig. S2).

**2.3.2 VOCs source apportionment and ozone formation potential**

Positive matrix factorization (PMF) model has been widely employed for VOCs source apportionment (Buzcu-Guven and Fraser, 2008; Leuchner and Rappenglück, 2010; Liu et al., 2016; Lyu et al., 2016). In this study, the EPA PMF 5.0 (US EPA, 2014) was employed and additional information was given in Appendix A.

The VOC concentrations are not proportional to the ozone formation potential due to their wide ranges of photochemical reactivity with OH radicals (Table 1). Two methods including propylene-equivalent concentrations (Propy-Equiv) and the maximum incremental reactivity (MIR) were adopted to analyze the ozone formation potential of VOCs. More details can be found in the research of Atkinson and Arey. (2003) and Zou et al. (2015).

In the photochemical and diurnal pattern discussion, the authors note that 8-14 is the time when the ratios are reduced because of photochemical removal due to enhancement of ambient temp. Ambient temp is not responsible for photochemical reaction. The increased in reactive radicals such as OH is. This should be corrected throughout this discussion.

Thanks for your correction, we have revised the description.

What are the uncertainties associated with authors calculations and discussion in section 3.5? This is important as the authors want to show such specific contribution sources in a region where the sources are so close to each other. Also, acetylene is a strong marker for combustion, but it has known to be released from some industrial and oil and gas sources too. It is OK to use is as such, but this point should be addressed in the paper.

The uncertainty (expressed by average ± standard deviation) of each source contribution was calculated based on the bootstrap technics in PMF analysis and was provided (annual contribution) in the revised manuscript. The "acetylene is released from some industrial and oil and gas sources too" is also mentioned in the revised manuscript.

This source was dominantly weighted by ethylene (95 ± 3.5 %), acetylene (97 ± 2.6%) and moderately influenced by BTEX. These species are key markers of combustion (Fujita, 2001; Watson et al., 2001; Jobson, 2004) or from petrochemical source (Brocco et al., 1997; Song et al., 2007). However, the independent combustion tracers such as $CO$, $NO_2$ and $PM_{2.5}$ were well correlated to this source contribution with Pearson correlation coefficients of 0.59, 0.49 and 0.77, respectively (Fig. 12c and Table S3). Therefore, this factor was attributed to combustion source..

The conclusion section should be changed to summary as it is just summarizing the results not adding any conclusions. The conclusions of the paper should relate only to this area. Also, when discussing all VOCs, are the benzene concentrations in this study higher than those in the middle of a high traffic urban area? Using the term VOC to encompass everything discussed here is not a correct comparison. Isn't BLH a meteorological condition? So, what is the difference between it and the met conditions affecting monthly and daily VOC variations? #2 in conclusions does not add anything and should be removed.

Thanks for your constructive comments and we changed the conclusion into summary. The benzene in this study (1.13 ± 1.62 ppbv) was lower than those measured in urban areas such as Mexico (1.96 ± 0.93 ppbv) (Garzón et al., 2015), Beijing (1.23 ± 0.68 ppbv) (Wang et al., 2010), Shanghai (1.81 ± 0.19 ppbv) (Cai et al., 2010), but was higher than that in Seoul (0.84 ± 0.72 ppbv) (Na and Kim, 2001) and Guangzhou (0.62 ppbv) (Zou et al., 2015). Yes, the BLH also belongs to meteorological condition and the BLH and other meteorological parameters (i.e., wind speed and temperature) both negatively correlated to VOCs variations. So, the meaningless #2 conclusion in original manuscript is removed as the referee suggested.

Reference

Cai, C., Geng, F., Tie, X., Yu, Q. and An, J.: Characteristics and source apportionment of VOCs measured in Shanghai, China, Atmos. Environ., 44(38), 5005–5014, doi:10.1016/j.atmosenv.2010.07.059, 2010.

Garzón, J. P., Huertas, J. I., Magaña, M., Huertas, M. E., Cárdenas, B., Watanabe, T., Maeda, T., Wakamatsu, S. and Blanco, S.: Volatile organic compounds in the atmosphere of Mexico City, Atmos. Environ., 119, 415–429, doi:10.1016/j.atmosenv.2015.08.014, 2015.

Na, K. and Kim, Y. P.: Seasonal characteristics of ambient volatile organic compounds in Seoul, Korea, Atmos. Environ., 35, 2603–2614, doi:10.1016/S1352-2310(00)00464-7, 2001.

Wang, B., Shao, M., Lu, S. H., Yuan, B., Zhao, Y., Wang, M., Zhang, S. Q. and Wu, D.: Variation of ambient non-methane hydrocarbons in Beijing city in summer 2008, Atmos. Chem. Phys., 10(13), 5911–5923, doi:10.5194/acp-10-5911-2010, 2010.

Zou, Y., Deng, X. J., Zhu, D., Gong, D. C., Wang, H., Li, F., Tan, H. B., Deng, T., Mai, B. R., Liu, X. T. and Wang, B. G.: Characteristics of 1 year of observational data of VOCs, NOx and O3 at a suburban site in Guangzhou, China, Atmos. Chem. Phys., 15(12), 6625–6636, doi:10.5194/acp-15-6625-2015, 2015.

The figures in this paper are not well prepared. There is too much going on in each figure (sometime irrelevant information). For example, for diurnal pattern figures the addition of day and night color only add information that are not useful.

Thanks for your thoughtful suggestions. The figures in this manuscript have been revised according to the referees' suggestion and are shown at the bottom of this response.

Specific comments:

Page 1, Line 15: The authors do not discuss the atmospheric behaviors of the VOCs in the paper. The closest they come to this is when they discuss the ozone formation potential of the VOCs. Most of the paper is on the VOC concentrations and correlations. The term behaviors should be deleted.

Thanks for your suggestion, the atmospheric behaviors have been deleted.

Page 1, Line 17-18: The authors note that the concentrations of VOCs in this study were 1-50 times higher than those measured in many other urban and industrial regions. This statement is vague. It should either be more specifically addressed (what VOCs, which other studies) or taken out.

Thanks for your suggestion, this sentence was deleted and re-organized. The revised statement is shown as following:

The average concentrations of total VOCs were 297 ± 372 ppbv and the main contributor was alkanes, accounting for 87.5% of the total VOCs.

Page 1, Line 21-22: After Section 3.3 is revised this sentence should be deleted. First, because the correlations were not very strong; second, because correlation (even if present) does not imply causality or influence.

Thanks for your suggestion, the section 3.3 has been revised, and this sentence has been deleted.

Page 1, Line 24: What do the authors mean by asphalt? Is this an asphalt making factory or just asphalt on the roads?

The asphalt means the asphalt making factory emissions and a black oil hill (made of natural asphalt) fugitives in the northwest of the sampling site.

Page 1, Line 25-26: What do the authors mean by "Clear temporal variations differed from one source to another was observed…" temporal variations of what?

In this sentence, we want to express that the diurnal variation of five different VOCs sources derived from the PMF model can be studied due to the high-resolution observation data and different VOCs sources exhibited different diurnal patterns. Therefore, this sentence is revised as following:

The five identified VOC sources exhibited various diurnal variation patterns due to their different emission patterns and the impact of meteorological parameters.

Page 1, Line 28: define CWT

The CWT (concentration weight trajectory) has been defined.

Page 1, Line 28: What do the authors mean by "local emissions"? Are these the emissions from the oil and gas field? From the urban location?

Yes, the local emission means the emission form both the oil and gas field and urban areas. Because in this study, we defined the local area as a circle and more descriptions can be found in the Appendix C.

Page 1, Line 29-30: Even though the data presented here is a good first step to look at the oil and gas emissions in this region, it is far from filling the gaps. Please revise this sentence to reflect what the paper actually does.

Thanks for your suggestion and this sentence has been revised.

Based on the high-resolution observation data, this study clearly described and analyzed the temporal variation of VOC emission characteristics at a typical oil and gas filed, which exhibited different atmospheric behaviors compared with that in urban and industrial areas.

Page 2, Line 5-6: The authors are implying that the reason for the detrimental effect of VOCs to human health is their activity as ozone precursors. This is not corrected. It is a part of the picture. VOCs such as benzene are carcinogens and bad for human health, independent of their activity as ozone precursors.

Thanks for your suggestions and we have revised this sentence.

As the key precursors of $O_3$ formation (Fujita, 2001; Geng et al., 2008; Ran et al., 2009; Lyu et al., 2016), different VOC categories exhibited different ozone formation potential (Carter, 1994; Atkinson and Arey, 2003; Zou et al., 2015). Some VOC species (i.e., benzene) exhibit detrimental effects on human health (Colman Lerner et al., 2012; He et al., 2015) and they have negative impacts on air quality (Vega et al., 2011).

Page 2, Line 7: Indicate the region of discussion because the previous studies in other regions have certainly focused on VOCs from oil and gas.

Yes, we have indicated the regions of the discussion.

Previous studies in China mainly focused on the measurements of VOCs in urban agglomerations such as the Pearl River Delta (PRD) region (Tang et al., 2007; Liu et al., 2008; Cheng et al., 2010; Ling et al., 2011), Yangtze River Delta (YRD) region (An et al., 2014; Li et al., 2016; Shao et al., 2016) and Beijing-Tianjin-Hebei (BTH) region (Li et al., 2015) and key megacities including Beijing (Song et al., 2007; Wang et al., 2010; Yuan et al., 2010), Shanghai (Cai et al., 2010; Wang, 2014), Guangzhou (Zou et al., 2015) and Wuhan (Lyu et al., 2016).

Page 2, Line 15-16: This sentence is not clear and needs revision. What do the authors mean by "depended on processing stages"? Are the authors discussing the processing stages (drilling, fracking, production) or something else?

In this sentence, we want to express that the VOC contributors in industrial and petrochemical areas are complex due to different process and atmospheric processes. The revised sentence is expressed as following:

A few studies were also conducted in industrial areas (An et al., 2014; Wei et al., 2015; Shao et al., 2016) and petrochemical industrial regions with a lot of VOC emissions (Lin et al., 2004; Wei et al., 2015; Jia et al., 2016; Mo et al., 2017). These studies found that the VOC sources and compositions are complex due to the different emission and atmospheric processes (Warneke et al., 2014)..

Page 2, Line 17-18: Gilman et al found that oil and gas emissions contribute strongly to some VOCs not all VOCs. Please correct this sentence to reflect that.

Thanks for your suggestion and this statement has been revised.

Gilman et al. (2013) found that oil and gas emissions strongly contribute to the propane and butanes in northeast Colorado.

Page 3, Line 9-10: How are the authors claiming that this specific study "contributes to establish the control measures of VOCs at this type of region"? What do they mean by this type of regions?

We conducted the VOCs source apportionment and calculated the contribution of each source to the ozone formation potential (OFP) in section 3.5. We found that despite the NG source contributed most (62.6%) to the ambient air VOCs, it contributed less to the ozone formation. However, the fuel evaporation contributed less to the VOCs (21.6%) but accounted for 41.9% of the OFP. Therefore, more attention should be paid to fuel evaporation due to its high ozone formation potential. That is why we are claiming that this study contributes to establish the control measures of VOCs at this type of region (refers to oil and gas resources rich areas).

Page 3, Line 15-16: Where did the authors get the production and deposit size data? Please cite source

The data source has been cited.

Page 4, Line 26: The authors note that 3-hour met data have been collected but then in Fig 2 they show hourly data? How was this done?

Due to our vague description, the meteorological data are indeed plotted using every three-hour data. Therefore, we modify the figure caption.

Page 4, Line 29: This should be in results not in methods.

Thanks, this sentence has been removed to results and discussions (section 3.3 temporal variation).

Page 10, Line 1-5: The difference is not only due to fuel evaporation. Based on the information presented here there is an urban area nearby and may contribute to the i/n pentane ratios. If this is not the case, the authors should add clarification.

We think that the urban sources such as vehicle emission contributed less to the $i$-pentane/$n$-pentane ratios. Firstly, the ratio of $i$-pentane/$n$-pentane in this study (1.03-1.24) was less that those reported emission ratios for vehicle emission or tunnel study (~2.2–3.8) (Conner et al., 1995; McGaughey et al., 2004). Secondly, the urban area is located the downwind direction of the sampling site (Fig. 1c), which less influences the sampling sites. Thirdly, the PMF analysis in section 3.5 indicated that the pentanes were from the NG and fuel evaporation. Therefore, we revised this part as following:

Additionally, $i$-pentane and $n$-pentane have similar physical and chemical characteristics (i.e., boiling point and reaction rate coefficients with hydroxyl radical), which result in less susceptible of the $i$-pentane / $n$-pentane ratio in source identification (Gilman et al., 2013). The pentanes are always form the NG emission, vehicle emission, liquid gasoline and fuel evaporation with the $i$-pentane/$n$-pentane ratios ranged between 0.82–0.89 (Gilman et al., 2010, 2013), ~2.2–3.8 (Conner et al., 1995; McGaughey et al., 2004), 1.5–3.0, and 1.8–4.6 (Watson et al., 2001),

respectively. As shown in Fig. 8b, the slopes of *i*-pentane / *n*-pentane were 1.03–1.24 in this study, suggested that the pentanes were more likely from the mixed sources of NG and fuel evaporation. This assumption was proved by the high loadings of pentanes in NG and fuel evaporation source compositions in section 3.5.5.

Page 10, Line 19-35: This section should be re-written for clarity.

Thanks for your suggestion, we have revised this part as following:

Generally speaking, when the reaction with OH radicals was the only factor controlling the seasonal ratio of longer atmospheric lifetime to shorter lifetime compounds (i.e., benzene / toluene, *m, p*-xylene / ethylbenzene), an increasing in ratio value from winter to summer would be expected (Russo et al., 2010). However, the seasonal variation of BTEX ratios in this study was opposite to the general behavior. For example, the benzene / toluene ratio decreased from winter-spring (0.63–0.69) to summer-fall (0.52–0.57) (Fig. 8c) and ethylbenzene / *m, p*-xylene ratio also decreased from autumn-winter (0.47–0.69) to spring-summer (0.19–0.37) (Fig. 8d). Same results were also observed both in industry areas (Miller et al., 2012) and urban areas (Ho et al., 2004; Hoque et al., 2008; Russo et al., 2010). The results obtained in this study indicated that there were other factors affecting the seasonal variation such as source emissions. The BTEX mainly originate from vehicle exhaust (Wang et al., 2010), solvent usage (Guo et al., 2004; Yuan et al., 2010) and petrochemical industry (Na and Kim, 2001; Hsieh et al., 2006; Baltrėnas et al., 2011). The ratio of *m, p*-xylenes / ethylbenzene here (2.2 ± 1.2) was within the ranges reported at a petrochemical area in southern Taiwan (1.5–2.6) (Hsieh et al., 2006) and the vicinity of a crude oil refinery at the Baltic region (3.0–4.0) (Baltrėnas et al., 2011). Therefore, the BTEX in this area was mainly from the oil refinery emission. The unexpected low ratios of benzene / toluene and *m, p*-xylene / ethylbenzene in summer was due to the strong oil refinery emission strength and this finding was verified by the seasonal source contribution results in section 3.5.1.

Page 11, Line 10: What is asphaltic?

Due to our vague description, we mean the asphalt and we corrected in the manuscript.

Examples of grammatical errors (these are not all the errors in the paper. The paper should go through a rigorous edit for grammar before publication).

Page 1, Line 13: add for between important and energy

Done

Page 1, Line 14: replace abundant with a grammatically correct adjective

We have replaced abundant with a lot of

The exploring, drilling, transportation and processing in oil and gas regions can release a lot of volatile organic compounds (VOCs).

Page 1, Line 17: revise "set of monitor system"

We have revised

Page 1, Line 25-26: revise for grammar

We have revised

The five identified VOC sources exhibited various diurnal variation patterns due to their different emission patterns and the impact of meteorological parameters.

Page 2, Line 2: revise "volcanos eruption"

Done

Page 2, Line 13: replace intensive with a grammatically correct adjective

We have replaced intensive with a lot of

A few studies were also conducted in industrial areas (An et al., 2014; Wei et al., 2015; Shao et al., 2016) and petrochemical industrial regions with a lot of VOC emissions

Page 3, Line 2: revise "few have concerned the local and regional source…"

We have revised this sentence

A few studies were also conducted in industrial areas (An et al., 2014; Wei et al., 2015; Shao et al., 2016) and petrochemical industrial regions with a lot of VOC emissions

Page 3, Line 3: replace long time with long-term

Done

Page 3, Line 5: Replace researches with grammatically correct term

Done

These practices mainly focus on the atmospheric fine particles (PM$_{2.5}$), few studies have concerned the local and regional source contributions of VOCs.

Page 3, Line 16: Please correct sentence so that "It" does not refer to 126 petrochemical plants but the area.

Done

This area can be divided into two regions with oil and gas operation and oil refinery at the north direction (Region 1) and petrochemical industry at the south direction (Region 2).

Page 3, Line 23: revise deeply

We have revised this sentence.

The study area is in the hinterland of Eurasia.

Page 3, Line 28: The authors make it sound like there were 56 monitoring sites. I assume this refers to the number of VOCs? Please revise sentence to reflect that.

We have revised this sentence.

From September 2014 to August 2015, 57 ambient VOCs designed as the O$_3$ precursors by the Photochemical Assessment Monitoring Stations (PAMS) were continuously sampled and measured using an online monitor system (TH-300B, Wuhan-Tianhong Instrument Co., Ltd, China) with two-hour time resolution.

Page 7, Line 13-17: Revise sentences There are much more, which made reading the manuscript very difficult. The authors should have used a technical writer.

We have revised this sentence.

High concentrations of alkanes, ethane and propane in ambient were also reported in other oil and natural gas operation and industrial areas in the US (Pétron et al., 2012; Helmig et al., 2014; Warneke et al., 2014). For instance, the average concentrations of ethane and propane were 74 ± 79 ppbv and 33 ± 33 ppbv, respectively in Horse pool and Uintah Basin in the winter of 2012. Despite the highly enhanced VOC levels were due to the temperature inversion, the VOC levels in Uintah Basin were still higher than those in the regional background areas as the existence of oil and gas exploitation activities (Helmig et al., 2014).

Thanks for your suggestion, we have carefully checked and corrected the grammar in the revised manuscript

Revised figures as the referees suggested.

[Figure]

**Figure 1.** The spatial distribution of oil gas bearing basins in China (a) and the terrain of the study area (b). The sampling site is about 11 km away from the urban area and located in the northeast of an oil refinery plant and southwest of an oil gas field. The northeasterly winds prevailed during the sampling periods (c)

(The legend (meaning of the different color) of Fig. 1a is provided. The filled color in Region 1 and 2 (Fig. 1b) is deleted to show the location of the sampling site)

[Figure]

**Figure 3**. Time series of the hourly concentrations (expressed in ppbv) of four categories of VOCs including alkanes (a), alkenes (b), acetylene (c), aromatics (d), and their fractions (e) during the sampling period.

(There is an error in Fig. 3e that the percentage of aromatics is not corrected and we have corrected it)

[Figure]

**Figure 4.** Comparison of the VOC concentrations (a), compositions (b) and the top five VOC species (c) in this study and former studies concerning the VOCs in the ambient of urban and industrial areas.

[a] Zou et al., (2015); [b] Cai et al., (2010); [c] Wang et al., (2010); [d] Garzón et al., (2015); [e] Baker et al., (2008); [f] Na and Kim, (2001); [g] Leuchner and Rappenglück, (2010); [h] Gilman et al., (2013); [i] Simpson et al., (2010); [j] Na et al., (2001); [k] An et al., (2014); [l] Shao et al., (2016)

(In order to give the data sources of the compared areas, we add the superscript in each area and the reference is provided)

[Figure]

**Figure 5.** Box and whisker plots of VOC profiles based on different scales during the whole sampling period. Box and Whisker plots are constructed according to 25th-75th and 5th -95th percentile of the calculation results.

(The irrelevant color is deleted.)

[Figure]

**Figure. 6** Seasonal and daily variations of ethane (a), ethylene (b), acetylene (c), and benzene (d) during the sampling period.

(The section 3.3 is discussing the temporal variations of different VOC species and Figure. 6 is revised to reflect the seasonal and daily variation.)

[Figure]

**Figure 7.** Diurnal variation of boundary layer height (BLH), VOCs, NO$_2$, and O$_3$ concentrations in different timescale: annual (a), winter (b) and summer (c). Solid line represents the average value and filled area indicates the 95th confidence intervals of the mean.

(The irrelevant color such as the day and night is deleted. We also add the diurnal variation of NO$_2$ in Fig. 7b and Fig.7c.)

[Figure]

**Figure 8.** Correlations (m=slope ± standard error (r²)) between compounds with similar atmospheric lifetimes including *i*-butane/*n*-butane (a) and *i*-pentane/*n*-pentane (b), and compounds with different lifetimes including benzene/toluene (c) and ethylbenzene/ *m, p*-xylenes (d).

(The x- and y- axis is exchanged (Fig. 8d) to show the ratio of ethylbenzene/*m, p*-Xylene (long lifetime compound/ short lifetime compound) as discussed in section 3.4)

[Figure]

**Figure 12.** Scatter plots of daily concentrations of trace gas and source contributions including oil refinery (a), NG (b), combustion (c), asphalt (d), and fuel evaporation (e) under different meteorological conditions (wind speed (WS), boundary layer height (BLH) and temperature (T)).

(The F1 (Oil refinery), F2 (NG), F3 (Combustion), F4 (Asphalt).and F5 (Fuel evaporation) were added in x-axis. In addition, the color maps of different meteorological data were unified.)

[Figure]

**Figure 13.** Diurnal variations of the contributions (expressed in ppbv) of five identified sources including oil refining process (a), NG (b), combustion source (c), asphalt (d) and fuel evaporation (e), and specific compounds with high loadings in each source profile. Note that the CO in combustion source was expressed in mg m⁻³.

(The irrelevant color (day and night) and the species that are not well correlated to the source contribution in are deleted. The full name of MCH (methylcyclohexane) is given. In addition, the layout of Fig. 13 is also changed)

[Figure]

**Figure 14**. Annual conditional probability function (CPF) plots of five identified VOCs source including oil refinery (a), NG (b), combustion (c), asphalt (d), and fuel evaporation (e).

(The irrelevant color is deleted and the layout is changed.)

---

## Author Comment (AC2) · 25 Jan 2018

**Response to Referee #4**

Thank you very much for your thoughtful and constructive comments on our manuscript. We have revised the manuscript accordingly. The detailed responses are given below point by point (in blue), and the revised manuscript is shown in red.

This paper entitled "One year monitoring of volatile organic compounds (VOCs) from an oil-gas station in northwest China" utilized a unique dataset to analyze the differences between the VOC concentrations, compositions, source contributions in an oil-gas station and other urban areas and industrials. The results seem to be interesting with unique characteristics of VOC compositions and sources in this kind of areas. Based on one-year online monitoring of VOC concentrations, the PMF model was successfully employed to source apportionment and the different timescale variations of different source contributions were discussed. The PSCF and CWT method were also employed to investigate the potential geographic origins of VOCs. A new method based on CWT was proposed to attempt to distinguish the local and regional contributions. I suggest this paper can be accepted after minor revision and addressing my questions.

We are very grateful to all important and helpful comments from the referee. The followings are our responses to each comment in detail.

The specific comments are listed as follows:

1. P1 Line 15 the sentence "the ambient VOCs from fifty-six Photochemical Assessment Monitoring Stations (PAMS) VOCs were continuously measured for an entire year (September 2014-August 2015) by a set of on-line monitor system from an oil-gas station in northwest China." confused me. Pls make it clear.

This sentence has been revised.

To understand the VOC levels, compositions and sources in such region, an oil and gas station in northwest China was chosen as the research site and fifty-seven VOCs designed as the photochemical precursors were continuously measured for an entire year (September 2014–August 2015) using an on-line monitoring system.

2. P1 Line 31: How about replace the keywords "source region and local-regional contribution" to local-regional contribution?

Thanks for your suggestion, and we have revised in the updated manuscript.

3. P2 Line 6: Insert references after "air quality."

We have added the reference

4. P2 Line 25 Insert references or link.

We have added the link, which the full-text of Air Pollution Prevention Control (APPC) can be found.

5. P4 Line 2~5: Please check and make sure the analysis method is correct.

The VOCs analysis method has been checked and some technic errors have been corrected.

Briefly, two-channels were installed to analyze VOCs separately. The water and carbon dioxide in the sampled air was firstly removed at a cold trap maintaining at −80 ℃ and then concentrated at −150 ℃ at another cold trap. After the purification and concentration, the VOCs were desorbed by rapid heating to 100 ℃. The $C_2$–$C_5$ VOCs were separated with a PLOT column (diameter: 0.32 mm, thickness of membrane: 1.5 μm, length: 60 m) and were quantified by the gas chromatograph-flame ionization detector (GC-FID, Agilent 7890). $C_5$–$C_{12}$ were separated by a DB-624 column (diameter: 0.25 mm, thickness of membrane: 3 μm and length: 60 m) and were quantified using mass spectrometer detector (MSD, Agilent 5975).

6. P4 Line 8. Technic errors. The PAMS standard gases contain 57 VOC species, including alkane (30), alkene (9), alkene (acetylene), and aromatic (17).

Thanks for your correction and we have revised it.

7. P6 Line 7: The author mentioned that the trajectories were mainly originated from the northwest during the whole sampling period. However, the wind rose (Fig. 1c) indicated the northeasterly winds prevailed in P5 Line 2. How to explain the difference?

The wind rose plot was drawn according the observation data at the sampling site, while the backward trajectory was analyzed using the National Center for Environmental Prediction's Global Data Assimilation System (GDAS) wind field re-analysis. In addition, the wind rose more reflects the instantaneous wind directions, while the back trajectories indicate the long-range transport in large spatial scale. The data source and spatial scale resulted the differences.

8. P7Line 15: You mean 33 ± 33ppbv?

Yes, we have corrected this error.

9. P9 Line 12: The concentrations of $O_3$ precursors decreased and $O_3$ increased?

This sentence is confusing and we have revised it in the manuscript. Actually, the $O_3$ precursors means VOCs and $NO_2$ in this study and we revised this sentence as below:

After sunrise, with the initiation of photochemical oxidation and the increasing of BLH, the concentrations of VOCs decreased while the $O_3$ increased rapidly.

10. P9 Line 16~23: When discuss the effects of BLH and photochemical reactions on VOC concentrations in summer and winter. I suggest more statistical method such as ANOVA analysis can be used to test the differences were significant or not.

Thanks for your suggestion. We conducted ANOVA analysis to discuss.

11. P11 Line 18-19 & P12 Line 8~10: The author compared the source contributions in different seasons using the percentage contributions (%) and volume contributions (ppbv) and it's paradoxical using both methods. For instance, in P12 Line 8~10, the percentage contribution in spring was the highest, however, the volume contribution was the lowest among the four seasons. How to explain or avoid?

Thanks for your reminder, we use the relative contribution (%) instead of both relative and absolute contribution (ppbv) to avoid the paradoxical expression.

12. P15 Line 18: Highest CPF values of oil refinery was found in the east direction (Fig. 14a, 14b, and 14d). This sentence confused me.

The CPF plot of this source was corresponding to Fig 14a, not referred to Fig. 14a, 14b, and 14d in the original manuscript. So, we have corrected this sentence.

The highest CPF value of oil refinery was found in east direction of the sampling site (Fig. 14a), which indicated the potential location of this source.

13. P25 Table 1: There are some mistakes such as an extra line under n-decane.

We have corrected this error.

14. P28 Table 3: I am wondering why the average value of four different seasons does not equal to annual value?

The average value of four different seasons does not equal to annual value which is due to the different calculation method. For instance, the local contribution of oil refining source in different seasons was calculated according to the raster analysis of each season, while the annual contributions were calculated according the whole year's CWT analysis. According to the equation 2, the $C_{bi}$ in different seasons was different. So, the average value of four seasons does not equal to the annual contribution.

15. P35 Figure 7 and P41 Figure 13: Due to the time resolution of meteorological parameters, BLH are three hours, while the time resolution of trace gases is one hour as the author mentioned. I suggest that the Pearson correlation can be conducted to give a more statistical reliable relation between VOC concentrations, different source contributions and trace gases.

Thanks for your constructive suggestion, the Pearson correlation between the VOC concentrations and different source contributions and trace gases were conducted. The species which were not well correlated with the corresponding source contributions were deleted. The revised manuscript is shown as following:

**3.5.1 Oil refinery**

The diurnal pattern of this source contribution was well correlated to the methylcyclohexane ($r = 0.76$, $p < 0.01$) and characterized by a double wave profile with the first peak at 02:00 LT and second peak at 06:00 LT (Fig. 13a).

**3.5.2 NG**

The diurnal variation of the NG leakage was significantly correlated ($p < 0.01$) with the diurnal pattern of propane $n$-butanes and $i$-butane with Pearson coefficients as 0.94, 0.87 and 091, respectively (Fig. 13b), which was also reported by Baudic et al. (2016).

**3.5.3 Combustion source**

The diurnal variation of combustion source was in accordance with the diurnal pattern of ethylene and CO with Pearson correlation coefficients as 0.71 ($p < 0.05$) and 0.84 ($p < 0.01$), respectively.

**3.5.4 Asphalt**

The diurnal variation of asphalt was different from other sources and well followed the diurnal patterns of decane ($r = 0.76$, $p < 0.01$) and undecane ($r = 0.86$, $p < 0.01$).

**3.5.5 Fuel evaporation**

On the contrary, the source contribution followed the diurnal variations of fuel evaporation tracers such as $i$-pentane, $n$-pentane and methycyclohexane, with Pearson correlation coefficients being 0.86 ($p < 0.01$), 0.87 ($p < 0.01$) and 0.67 ($p < 0.05$), respectively.

---

## Author Comment (AC3) · 25 Jan 2018

**Response to Referee #3**

Thank you very much for your thoughtful and constructive comments on our manuscript. We have revised the manuscript accordingly. The detailed responses are given below point by point (in blue), and the revised manuscript is shown in red.

This paper describes one-year continuous monitoring of VOCs around an oil-gas region in northwest China in order to clarify atmospheric behavior of VOCs in such region. The authors revealed temporal variations such as seasonal and diurnal variations of VOCs around the oil-gas region and analyzed factors of such variations. In addition, they performed source analyses of VOCs and discussed source of VOCs in this region quantitatively.

General comments:

As the authors mentioned, VOCs are main precursors of tropospheric ozone and it is important to clarify atmospheric behavior of VOCs. Examples of VOC observations in oil-gas regions are low, especially; there are few continuous observations of VOCs with high time resolution. The authors supply valuable data and information. In addition, the authors conducted quantitative source analyses of VOCs. I recommend this paper to be published in Atmospheric Chemistry and Physics.

However, I found several dubious points in this paper. The authors should revise appropriately.

The authors would like to thank the reviewer for the detailed comments, which help to improve the manuscript. We have tried to clarify the points raised by the reviewer and to answer all remarks.

Specific comments:

The authors performed several discussions using $NO_2$. Why do the authors use $NO_2$ instead of $NO_x$? I think it is preferable to use $NO_x$ instead of $NO_2$ (or both $NO_2$ and $NO_x$) for many of such discussions. The authors would observe NO and $NO_2$ because they used a TEI $NO_x$ analyzer based on a chemiluminescence method.

Thanks for your suggestion that using $NO_x$ instead of $NO_2$ to discuss, however, the $NO_2$ and other air pollutants data were from the Qingyue Open Environmental Data Center, which only $NO_2$ is available. So, we had to use the $NO_2$ to discuss and we will notice this point in our further study.

$NO_2$ and $NO_x$ concentrations measured by a TEI $NO_x$ analyzer are not accurate because of interferences of descendant spices of $NO_x$ such as $HNO_3$ and PANs. The authors should evaluate such interferences. Especially, organic nitrates could interfere the values of $NO_2$ concentrations obtained by a TEI $NO_x$ analyzer under high concentrations of large hydrocarbons.

Thanks for your comments. There is a technic error in the manuscript that the $NO_2$ was actually measured using the automated monitors (TH-2000 series, Wuhan-Tianhong Instrument Co., Ltd, China), which also determine the $NO_x$ using the chemiluminescence technic. Based on the chemiluminescence method, the measured NO concentrations is accurate. A molybdenum converter was used for $NO_2$ measurement, and as a result, part of the $NO_y$ (e.g., peroxyacetylnitrate (PAN), $HNO_3$, and alkyl nitrates) may have been transformed to $NO_2$ during the sampling.  Therefore, a method developed by Lamsal et al. (2008) was used to correct our $NO_2$ data according to the following formula:

$$CF = \frac{NO_2}{NO_2 + \sum AN + (0.95PAN) + (0.35HNO_3)}$$

where CF is the correct factor, $\sum AN$ is the alkyl nitrates. However, in this study, we do not measure the PAN and $HNO_3$ and simulate the alkyl nitrates concentrations. Therefore, the $NO_2$ concentrations discussed is this study were considered greater than the actual values (Dunlea et al., 2007; Zou et al., 2015) and we used the average value of $NO_2$ concentration to discuss. We will notice this issue in further study.

Reference

Dunlea, E. J., Herndon, S. C., Nelson, D. D., Volkamer, R. M., San Martini, F., Sheehy, P. M., Zahniser, M. S., Shorter, J. H., Wormhoudt, J. C., Lamb, B. K., Allwine, E. J., Gaffney, J. S., Marley, N. A., Grutter, M., Marquez,

C., Blanco, S., Cardenas, B., Retama, A., Ramos Villegas, C. R., Kolb, C. E., Molina, L. T. and Molina, M. J.: Evaluation of nitrogen dioxide chemiluminescence monitors in a polluted urban environment, Atmos. Chem. Phys., 7(10), 2691–2704, doi:10.5194/acp-7-2691-2007, 2007.

Lamsal, L. N., Martin, R. V., van Donkelaar, A., Steinbacher, M., Celarier, E. A., Bucsela, E., Dunlea, E. J. and Pinto, J. P.: Ground-level nitrogen dioxide concentrations inferred from the satellite-borne Ozone Monitoring Instrument, Journal of Geophysical Research, 113(D16), doi:10.1029/2007JD009235, 2008.

On page 9, lines 6-7, "It should be noted that VOCs: : : as well as BLH.": I think $NO_2$ concentrations are controlled solar UV and concentrations of NO and $O_3$ as well as BLH, but are VOCs controlled concentrations of NO and $O_3$? (I don't think so.) The authors should discuss this matter separating VOCs and $NO_2$.

Thanks for your comment. We have revised this part.

The VOCs had a reverse trend with $O_3$ ($r = -0.82$, $p < 0.01$). The lower BLH and less photochemical activities resulted in peak values for VOCs and low $O_3$ concentrations before sunrise (6:00 local time). After sunrise, with the initiation of photochemical oxidation and the increasing of BLH, the concentrations of VOCs decreased while the $O_3$ increased rapidly. The minimum of VOCs and occurred at about 12:00–14:00 LT was resulted from both dispersion or dilution conditions and photochemical reactions (with highest $O_3$ concentrations at l4:00 LT) in the afternoon. The diurnal variation of $NO_2$ was controlled by BLH, $O_3$ and photochemical reactions (i.e., OH radical) and showed a double peak.

Table 1: The authors should explain $r^2$.

Done.

---

## Author Comment (AC4) · 25 Jan 2018

**Response to Referee #2**

Thank you very much for your thoughtful and constructive comments on our manuscript. We have revised the manuscript accordingly. The detailed responses are given below point by point (in blue), and the revised manuscript is shown in red.

This paper reported one year VOC measurements in high frequency at oil-gas station area in China. High frequency VOC data enable to show diurnal variations of VOCs in the sampling site and strong influence of meteorological condition for VOC concentrations was elucidated. Using PMF analysis, VOC emission was successfully categorized into five sources. Further, CPF analysis gave more information about the direction of the VOC emission sources. Also, backward trajectory based analysis (PSCF, CWT) were applied.

[General comments]

Similar VOCs measurement and analysis were already demonstrated in urban area, but the detailed VOC observation at oil-gas station area will be important in view point of source area information. Because strong VOC emission sources are close to the measurement site, the analyzed results would be simpler and seems to be reasonable. But I am not sure about the validity of PSCF and CWT analysis in this measurement. When there is a strong source very close to the measurement site (like 2 km from Oil refinery and 6km from Oil-gas field in Fig 1c), the backward trajectories passed the near emission source will be counted as high concentration over its whole trajectory rout. Therefore, the raster analysis (distinguish local and regional area) would be reasonable trial. (But I am afraid that the influence of very close emission sources cannot be excluded, because that 12h radius will be still long and concentration of each grid was estimated from CWT.)

We thank the reviewer for the valuable comments. All of them have been addressed in the revised manuscript. Please see our itemized responses below.

Explanation for the validity of PSCF and CWT

Despite the potential source contribution function (PSCF) method has deficiencies such as the determination of the statistical significance of its outcome is difficult (Stohl, 1996). The PSCF (Polissar et al., 1999; Hsu et al., 2003) and concentration weighted trajectory (CWT) are frequently used to figure out the direction and sources of air pollution at a receptor site. To reduce the uncertainty and increase the confidence of the results, a weighted function is widely used (Wang et al., 2016). Grid cells for which high PSCF values are calculated from the arrival of air parcels at a receptor site with pollutant concentrations higher than a given value. In this study, the 75th percentiles of each source contribution were set as the criterion (Bressi et al., 2014; Wang et al., 2015). The detailed information about the PSCF and CWT calculation can be found in Appendix B.

The PSCF results represent potential source directions rather than locations because PSCF modeling evenly distributes weight along the path of trajectories as the referee mentioned. This even weighting results in a trailing effect so that areas upwind and downwind of the sources are likely to be identified as sources as well (Hsu et al., 2003). Therefore, in this study, we simply define a circle with 12 h radius area as the local areas and the outer areas of this circle is regional. The reasons are listed below:

(1) The duration of m, p-xylene decreasing from highest concentrations (at 02:00 LT) to lowest (14:00 LT) was 12 h (section 3.4). The atmospheric lifetime of *m, p*-xylene is about 11.8 h assuming that the OH radical equals to $10^6$ rad $cm^{-3}$. The compounds with atmospheric lifetime longer than *m, p*-xylene can be transported from long distance or accumulated in the local area.

(2) The endpoint of every 2 h backward trajectories in the first 24 h was tested to find the optimum range of "local and nearby" area (Fig. C1). As the backward time increasing from 2 h to 24 h, the area covered by the long air masses increased significantly. Before the first 5 h, the air masses were mainly from the northwest and the east of the sampling site. From 7 h to 12 h, air masses from the northeast, southeast and southwest reached the receptor site and

the "shape of local area" formed. After 12 h, the air masses, especially for the trajectories from the west transported for long distance reached the sampling site, indicating the regional conditions.

As the referee mentioned, the 12 h radius to define the local and regional transport was still long that the very close emission sources cannot be excluded. The method used in this study was a trial to give the quantitative information of the local and regional transport despite flaws existed.

Reference

Bressi, M., Sciare, J., Ghersi, V., Mihalopoulos, N., Petit, J.-E., Nicolas, J. B., Moukhtar, S., Rosso, A., Féron, A., Bonnaire, N., Poulakis, E. and Theodosi, C.: Sources and geographical origins of fine aerosols in Paris (France), Atmos. Chem. Phys., 14(16), 8813–8839, doi:10.5194/acp-14-8813-2014, 2014.

Hsu, Y.-K., Holsen, T. M. and Hopke, P. K.: Comparison of hybrid receptor models to locate PCB sources in Chicago, Atmos. Environ, 37(4), 545–562, doi:10.1016/S1352-2310(02)00886-5, 2003.

Stohl, A.: Trajectory statistics-A new method to establish source-receptor relationships of air pollutants and its application to the transport of particulate sulfate in Europe, Atmos. Environ, 30(4), 579–587, doi:10.1016/1352-2310(95)00314-2, 1996.

Wang, L., Liu, Z., Sun, Y., Ji, D. and Wang, Y.: Long-range transport and regional sources of $PM_{2.5}$ in Beijing based on long-term observations from 2005 to 2010, Atmos. Res, 157, 37–48, doi:10.1016/j.atmosres.2014.12.003, 2015.

Wang, Q., Liu, M., Yu, Y. and Li, Y.: Characterization and source apportionment of $PM_{2.5}$-bound polycyclic aromatic hydrocarbons from Shanghai city, China, Environ. Pollut, 218, 118–128, doi:10.1016/j.envpol.2016.08.037, 2016.

[Detailed comments]

Fig. 1: What is 2 type green area in Fig 1(a)? What is blue area in Fig1(b)? Exact location of the sampling site is difficult to see in Fig1 (b) (because of blue color). Is the sampling site located in Region 1? What is the rose figure

in Fig1c? Wind speed or VOC concentration from each direction? If this rose figure shows wind direction, figure

caption will be incorrect ("Northwesterly and northeasterly wind prevailed").

Fig. 1 has been revised. Fig. 1a shows the spatial distribution of oil and gas bearing basins in China in different colors

(yellow: gas well, red: oil well, and green: depositional basin). In the Fig. 1b, the sampling site is located in region 1

Fig. 1c shows the wind rose and the prevailing wind.

[Figure]

**Figure 1.** The spatial distribution of oil gas bearing basins in China (a) and the terrain of the study area (b). The sampling site is about 11 km away from the urban area and located in the northeast of an oil refinery plant and southwest of an oil gas field. The northeasterly winds prevailed during the sampling periods (c)

Page4 L3: PLOT column will be used for $C_2$-$C_4$ separation in GC. For $C_2$-$C_4$ trap, some absorbent (tenax etc) would

be used. Please check the explanation about VOC measurement system.

The VOCs analysis method has been checked and some technic errors have been corrected.

Briefly, two-channels were installed to analyze VOCs separately. The water and carbon dioxide in the sampled air was firstly removed at a cold trap maintaining at $-80$ ℃ and then concentrated at $-150$ ℃ at another cold trap. After the purification and concentration, the VOCs were desorbed by rapid heating to $100$ ℃. The $C_2$–$C_5$ VOCs were separated with a PLOT column (diameter: 0.32 mm, thickness of membrane: 1.5 μm, length: 60 m) and were quantified by the gas chromatograph-flame ionization detector (GC-FID, Agilent 7890). $C_5$–$C_{12}$ were separated by a DB-624 column (diameter: 0.25 mm, thickness of membrane: 3 μm and length: 60 m) and were quantified using mass spectrometer detector (MSD, Agilent 5975).

Table 1: Please explain "MDL" (method detection limit). "KOH" will be better to write in small characters.

Thanks for your suggestion and we have revised it.

Fig3: Is the colors in Fig. 3(e) same as Fig. 3(a)-(d)? (In Fig. 3(e), blue (acetylene?) seems larger contribution.)

Thanks for your reminder, the color in Fig. 3(e) is the same as Fig. 3(a)-(d) and we find some errors in Fig. 3e and therefore it has been revised. After correction, the contribution of aromatics (magenta) is greater than acetylene (blue).

[Figure]

**Figure 3.** Time series of the hourly concentrations (expressed in ppbv) of four categories of VOCs including alkanes (a), alkenes (b), acetylene (c), aromatics (d), and their fractions (e) during the sampling period.

Fig. 6 and text: Are $O_3$, BLH etc 24-hour average or only daytime average? There are strong diurnal variations in BLH, $O_3$ and VOCs, so these plots would be different when you use whole day or only daytime data.

Yes, the O₃, BLH etc. are 24-hour average in the original Fig. 6. In this part, we want to discuss the temporal variation of VOCs in different timescales. For daily variation patterns (Fig. 6), we used the 24-h average values to discuss. Of course, we agree with the referee's opinion that the concentrations of $O_3$, VOC and $NO_2$ concentrations and BLH would be different during the daytime and nighttime. So, we discussed the diurnal variation of these factors using the 95% confidence interval of gases (VOCs, $O_3$ and $NO_2$) and average ± standard deviation of BLH as shown in Fig. 7. Therefore, we revised the Fig. 6 and Fig. 7 as below:

[Figure]

**Figure. 6** Seasonal and daily variations of ethane (a), ethylene (b), acetylene (c), and benzene (d) during the sampling period.

[Figure]

**Figure 7.** Diurnal variation of boundary layer height (BLH), VOCs, $NO_2$, and $O_3$ concentrations in different timescale: annual (a), winter (b) and summer (c). Solid line represents the average value and filled area indicates the 95th confidence intervals of the mean.

Fig12 (d): Is it correct "F1-F4" ? (Is it "CO-F4" ?)

Yes, it is corrected as "F1-F4" in Fig. 12d. To avoid the vague description, the Fig. 12 was revised. For instance, the F1 (Oil refinery), F2 (NG), F3 (Combustion), F4 (Asphalt) and F5 (Fuel evaporation) were added in x-axis. In addition, the color maps of different metrological data were unified.

[Figure]

Figure 12. Scatter plots of daily concentrations of trace gas and source contributions including oil refinery (a), NG (b), combustion (c), asphalt (d), and fuel evaporation (e) under different meteorological conditions (wind speed (WS), boundary layer height (BLH) and temperature (T))

Fig13: Please explain MCH and MCP. In the right axis of (a), (b), (e), "/" will be better to show as ",". They are not ratio, but just concentration. "Cyclohexane" etc would be better to show as dot and line.

Thanks for your suggestion. This figure has been revised.

[Figure]

**Figure 13.** Diurnal variations of the contributions (expressed in ppbv) of five identified sources including oil refining process (a), NG (b), combustion source (c), asphalt (d) and fuel evaporation (e), and specific compounds with high loadings in each source profile. Note that the CO in combustion source was expressed in mg m$^{-3}$

P12 L24-2: Diurnal pattern of acetylene seems to be different. (Acetylene did not decrease during daytime.)

Yes, we agree with the referee's idea and this expression has been revised.

The diurnal variation of combustion source was in accordance with the diurnal pattern of ethylene and CO with Pearson correlation coefficients as 0.71 ($p < 0.05$) and 0.84 ($p < 0.01$), respectively.

P15 L20-22: "Northeast to Southwest" Is this "Southeast to Southwest"?

Yes, thanks for your correction.

P15 L23-24: Fig.14(b), highest peak is in east direction.

Thanks for your correction, we also find that there were some grids with high PSCF value in the east direction of the sampling site. So, we have revised this sentence as following:

Indeed, high values of CPF, PSCF and CWT were found in the east direction (Fig. 14b, Fig. 15b and Fig. 16b), which indicated that the potential geographic origins of NG.

---

## Author Response (AR2)

Dear Prof. Kimitaka Kawamura and referees:

Thank you very much for your thoughtful and constructive comments on our manuscript. We have revised the manuscript accordingly. The detailed responses are given below point by point (in blue).

**Response to Referee #1**

The authors have improved the paper based on the referee recommendations. The paper looks great and much more coherent. There were some minor technical errors, some of which are listed below:

The page and line numbers are based on the author response document.

Many thanks for your help in improving the manuscript. The comments have been answered and corresponding corrections have been done.

Page 33, Line 16: designed should be designated

Thanks for your correction and we have corrected it.

Page 33, Line 29: can delete variation

We have deleted it.

Page 34, Line 3: filed should be field

We have corrected it.

Page 34, Line 4: Please clarify what you mean by atmospheric behavior

The atmospheric behavior actually means the VOC levels, compositions, and origins in this study. We have clarify it in the revised manuscript.

Page 34, Line 28: When authors talk about "these studies" do they mean the studies they noted before? If yes, then the Warneke citation is confusing. If no, then please reword.

Yes, "these studies" refers to studies noted before. In this paragraph, we mentioned two times "these studies". The first refers the studies conducted in urban areas, the second means the studies conducted in petrochemical industrial areas. To avoid confusing, we have corrected it as following:

These studies carried out in industrial areas found that the VOC sources and compositions are complex due to the different emission and atmospheric processes.

Page 42, Line 29: Do you mean met variables or VOCs by "these species". Please clarify.

These species mean the VOC species (ethene, ethylene, acetylene, and benzene) and we have corrected it.

Page 42, line 29: should be were observed in winter

We have corrected it.

Page 42, line 30: was should be were

We have corrected it.

Page 43, line 4: less should be reduced

We have corrected it.

Page 43, line 5: proved should be shown or something similar

Thanks for your correction and we have revised it.

Page 43, line 6: may be also should be may also be

We have corrected it.

Page 43, line 26: delete and

We have corrected it.

Page 43, line 34: sentence should be revised to correct grammar

We have revised it as following:

However, the VOC concentrations of summer and winter decreased by 8.3 times and 2.3 times, respectively, from their maximum to the minimum.

Table S2: has a footnote about one * but there are not numbers with one * indicated in the table.

Thanks for your correction and we have corrected it.

The Y axis for Figure 6 b, c, and, d are incorrect.

We have corrected the Y axis of Fig. 6.

**Response to Referee #2**

The paper reported the high frequent VOC measurements in oil, gas supplying region for 1 year. These detailed VOC measurements will tell important information of sources and atmospheric chemical reactions and air mixing. Several analysis were applied to explain the observed results. Some of the analysis seems to be not suitable for interpretation of the observed results at this site. But they will also be good trial to explain the observed VOC results.

There are some mistakes in figures.

Thanks for your comments and suggestions on the manuscript improvement. The mistakes in some figures have been corrected.

Fig 1 : What indicate the color of the wind rose figures? Wind speed?

The color in wind rose indicates the wind speed and we have revised this figure.

Fig4(a) : Please add other value in addition to 100ppbv in x axis (large scale). Otherwise, we can not read the interval of the x axis.

Thanks for your suggestion and we have revised it.

Fig6 (b), (d) : Legends of Y axis will be ethylene or benzene.

We have corrected it.

Fig13 (c) : Right axis would be "ethylene".

Thanks for your correction and we have revised it.

**Response to Referee #3**

Accepted as is

Many thanks for your valuable comments.

[revised manuscript text omitted]